



**Quantifying spatiotemporal variability in zooplankton dynamics in the Gulf of Mexico with**
**a physical-biogeochemical model**
Taylor A Shropshire[1,2], Steven L Morey[3], Eric P Chassignet[1,2], Alexandra Bozec[1,2], Victoria J
Coles[4], Michael R Landry[5], Rasmus Swalethorp[5], Glenn Zapfe[6], Michael R Stukel[1,2]
[1]Department of Earth Ocean and Atmospheric Sciences, Florida State University, Tallahassee, FL 32303
[2]Center for Ocean-Atmospheric Prediction Studies, Florida State University, Tallahassee, FL
[3]School of the Environment, Florida A&M University, Tallahassee, FL
[4]University of Maryland Center for Environmental Science, PO Box 775 Cambridge MD 21613
[5]Integrative Oceanography Division, Scripps Institution of Oceanography, 8622 Kennel Way, La Jolla, CA 92037
[6]University of Southern Mississippi, Division of Coastal Sciences, Hattiesburg, MS, 39406
Correspondence: Taylor A. Shropshire (tshropshire@fsu.edu)



**Abstract**

Zooplankton play an important role in global biogeochemistry and their secondary production supports valuable fisheries of the world's oceans. Currently, zooplankton abundances cannot be estimated using remote sensing techniques. Hence, coupled physical-biogeochemical models (PBMs) provide an important tool for studying zooplankton on regional and global scales. However, evaluating the accuracy of zooplankton abundance estimates from PBMs has been a major challenge as a result of sparse observations. In this study, we configure a PBM for the Gulf of Mexico (GoM) from 1993-2012 and validate the model against an extensive combination of in situ biomass and rate measurements including total mesozooplankton biomass, size-fractionated mesozooplankton biomass and grazing rates, microzooplankton specific grazing rates, surface chlorophyll, deep chlorophyll maximum depth, phytoplankton specific growth rates, and net primary production. Spatial variability in mesozooplankton biomass climatology observed in a multi-decadal database for the northern GoM is well resolved by the model with a statistically significant ($p < 0.01$) correlation of 0.90. Mesozooplankton secondary production for the region averaged $66 \pm 8$ mt C yr$^{-1}$ equivalent to approximately 10% of NPP and ranged from 51 to 82 mt C yr$^{-1}$. In terms of diet, model results from the shelf regions suggest that herbivory is the dominant feeding mode for small mesozooplankton (<1-mm) whereas larger mesozooplankton are primarily carnivorous. However, in open-ocean, oligotrophic regions, both groups of mesozooplankton have proportionally greater reliance on heterotrophic protists as a food source. This highlights the important role of microbial and protistan food webs in sustaining mesozooplankton biomass in the GoM which serves as the primary food source for early life stages of many commercially-important fish species, including tuna.



## 1.     Introduction

Within marine pelagic ecosystems zooplankton function as an important energy pathway between the base of the food chain and higher trophic levels such as fish, birds, and mammals (Landry et al., 2019; Mitra et al., 2014). Zooplankton also have a well-documented impact on chemical cycling in the ocean (Buitenhuis et al., 2006; Steinberg and Landry, 2017; Turner, 2015).The ecological roles of zooplankton, however, are varied and taxon-dependent. Globally, protistan grazing is the largest source of phytoplankton mortality, accounting for 67% of daily phytoplankton growth (Landry and Calbet, 2004). Protistan zooplankton function primarily within the microbial loop leading to efficient nutrient regeneration in the surface ocean (Sherr and Sherr, 2002; Strom et al., 1997). By contrast, mesozooplankton contribute significantly less to phytoplankton grazing pressure consuming an estimated 12% of primary production (PP) globally (Calbet, 2001) yet strongly impact the biological carbon pump. In addition to top-down grazing pressure on phytoplankton, mesozooplankton impact the biological carbon pump through production of sinking fecal pellets, consumption of sinking particles and active carbon transport during diel vertical migration (Steinberg and Landry, 2017; Turner, 2015). While contributing notably less to phytoplankton grazing pressure than protists, herbivorous mesozooplankton are important to study as they are often associated with shorter food chains that enable efficient energy transfer from primary producers to higher trophic levels of particular societal interest such as economically valuable fish species and/or their planktonic larvae.

Zooplankton populations have been identified as being vulnerable to impacts of a warming ocean (Caron and Hutchins, 2013; Pörtner and Farrell, 2008; Straile, 1997), through both impacts of temperature on metabolic rates (Ikeda et al., 2001; Kjellerup et al., 2012) and thermal stratification-driven alterations in food web structure (Landry et al., 2019; Richardson, 2008). Studies aimed at monitoring and predicting zooplankton populations are therefore critical to understanding the first-order effects of a warming ocean on marine ecosystems given the importance of secondary production and the impact zooplankton have on biogeochemical cycling. Despite their importance, historically zooplankton have been sampled with limited temporal and spatial resolution. While remote sensing has provided an enormous advancement in observing ocean hydrodynamics and phytoplankton variability, zooplankton abundance cannot currently be estimated from space. Thus numerical models provide a unique oceanographic research tool for studying zooplankton on basin and global scales (Buitenhuis et al., 2006; Sailley et al., 2013; Werner et al., 2007). Evaluating the



accuracy of zooplankton abundance estimates from numerical models, such as three-dimensional
physical-biogeochemical ocean models (PBMs), has been a major challenge in previous modeling
studies as a result of sparse ship-based observations in most regions (Everett et al., 2017).
Consequently, zooplankton dynamics have been under studied and under validated in PBMs.
Instead, PBMs are typically validated predominately against surface chlorophyll (Chl) from
remote sensing (Doney et al., 2009; Gregg et al., 2003; Xue et al., 2013).
In most marine environments, phytoplankton net growth rates and hence biomass are determined
primarily by the imbalance between phytoplankton growth and zooplankton grazing (Landry et
al., 2009). PBMs can accurately predict phytoplankton standing stock (i.e. compare well with
satellite Chl observations) despite being driven by the wrong underlying dynamics leading to major
errors in model estimates of secondary production and nutrient cycling (Anderson, 2005; Franks,
2009). For instance, parameter tuning using only surface Chl as a validation metric can allow broad
patterns in phytoplankton biomass to be reproduced even with gross over- or underestimation of
phytoplankton turnover times.  Similarly, even a model that is validated against satellite Chl and
net primary production might completely misrepresent the proportion of phytoplankton mortality
mediated by zooplankton groups, leading to inaccurate estimates of secondary production. Hence,
validating PBMs against zooplankton dynamics is key to increasing confidence in model solutions.
The importance of this validation is further witnessed when considering the impact zooplankton
have on the behavior of biogeochemical models (Everett et al., 2017). Differences in simulated
zooplankton communities expressed through the number of functional types, various mathematical
grazing functional responses, and the arrangement of transfer linkages have been shown to have
substantial impacts on simple and complex biogeochemical model solutions (Gentleman et al.,
2003; Gentleman and Neuheimer, 2008; Mitra et al., 2014; Murray and Parslow, 1999; Sailley et
al., 2013).
The Gulf of Mexico (GoM) is a particularly suitable study region for examining zooplankton
dynamics with PBMs. In the northern and central Gulf, zooplankton abundance has been
extensively measured for over three decades (1982-present) by the Southeast Area Monitoring and
Assessment Program (SEAMAP). Within the SEAMAP dataset, measured zooplankton abundance
exhibits strong spatiotemporal variability, due to complex physical circulation features within the
GoM. The circulation in regions off the shelf is characterized by substantial upper layer mesoscale





activity driven primarily by the energetic Loop Current (Forristall et al., 1992; Maul and Vukovich,
1993; Oey et al., 2005). In contrast, coastal and shelf circulation patterns are predominantly wind-
driven (Morey et al., 2003a, 2013). Freshwater discharged by the Mississippi River and other
smaller rivers is frequently entrained offshore by shelf break interaction with mesoscale features
(e.g., anti-cyclonic loop current eddies), leading to strong horizontal and vertical gradients in
physical and biogeochemical quantities (Morey et al., 2003b). These gradients overlap with the
SEAMAP study region resulting in zooplankton biomass sample collection across
biogeochemically heterogeneous and "patchy" environments which provides a powerful model
constraint. For instance, Chl can range across approximately three orders-of-magnitude (~0.01 –
10 mg Chl $m^{-3}$) from oligotrophic to eutrophic waters. Similarly, mesozooplankton ($\geq$ 202 µm)
biomass is highly variable ranging from 0.1 – 160 mg C $m^{-3}$ in the SEAMAP dataset.
Several PBM studies have been conducted in the GoM, all primarily examining nutrient and
phytoplankton dynamics. Early work by Fennel et al. (2011) examined phytoplankton dynamics
on the Louisiana and Texas continental shelf, concluding that loss terms (e.g., grazing) rather than
growth rates dictated accumulation rates of phytoplankton biomass. With the same biogeochemical
model, Xue et al. (2013) conducted the first gulf-wide PBM study to investigate broad seasonal
biogeochemical variability and used the model to constrain a nitrogen budget for the shelf. More
recently, Gomez et al. (2018) implemented a biogeochemical model with multiple phytoplankton
and zooplankton functional types to gain a more detailed understanding of nutrient limitation and
phytoplankton dynamics in the GoM. To examine phytoplankton seasonality and biogeography in
the oligotrophic Gulf, Damien et al. (2018) validated a PBM based on a unique subsurface
autonomous glider dataset. Together, these studies have demonstrated the utility of PBMs for
investigating the GoM lower trophic level and have also highlighted the key role zooplankton play
in the ecosystem. Specifically, both Fennel et al. (2011) and Gomez et al. (2018) identified the
importance of zooplankton in modulating the simulated seasonal patterns of phytoplankton
biomass, emphasizing the importance of top-down control on the shelf. Although results on the
simulated zooplankton community were not presented, Damien et al. (2018) noted that biotic
processes such as grazing pressure, are "essential to fully understanding the functioning of the
GoM ecosystem." However, in these studies zooplankton validation is largely absent.



In this study, we configured a PBM for the GoM to estimate zooplankton abundance and analyze
zooplankton community dynamics. The PBM is forced by three-dimensional hydrodynamic fields
from a data assimilative Hybrid Coordinate Ocean Model (HYCOM) hindcast of the GoM
(http://www.hycom.org). The PBM is based on the biogeochemical model NEMURO (North
Pacific Ecosystem Model for Understanding Regional Oceanography; Kishi et al., 2007), which is
substantially modified here for application to the GoM. The model is integrated over 20-years
(1993-2012) and validated extensively against a combination of remote and in situ measurements
including total mesozooplankton biomass, size-fractionated mesozooplankton biomass and
grazing rates,  microzooplankton specific grazing rates, surface Chl, deep Chl maximum depth,
phytoplankton specific growth rates, and net primary production. The goals of this study were to:
1) develop and validate a PBM to estimate mesozooplankton abundance in the GoM, 2)
characterize the spatiotemporal variability in mesozooplankton dietary composition, and 3)
quantify regional mesozooplankton secondary production. We focus primarily on the oligotrophic,
open ocean GoM where prey (i.e. zooplankton) availability may be limiting for fish, their larvae,
and other higher trophic levels.
**2        Methods and data**
**2.1       Ocean model framework**
**2.1.1     Biogeochemical model description**
The biogeochemical model for this study is based on NEMURO (Kishi et al., 2007) but has been
modified and parameterized to more accurately reflect the ecology of the GoM. NEMURO is a
concentration-based lower trophic level ecosystem model originally developed and parameterized
for the North Pacific. Like most marine biogeochemical models, it is structured around simplified
representations of the lower food web originating from earlier nutrient-phytoplankton-zooplankton
models (Fasham et al., 1990; Franks, 2002; Riley, 1946; Steele and Frost, 1977). Complexity is
added through additional state variables and transfer functions with the specific goal of resolving
dynamics within the nutrient, phytoplankton, and zooplankton pools. In total, NEMURO has
eleven state variables: six non-living state variables – nitrate ($NO_3$), ammonium ($NH_4$), dissolved
organic nitrogen (DON), particulate organic nitrogen (PON), silicic acid ($Si(OH)_4$), and particulate
silica (Opal); two phytoplankton state variables – small (SP) and large phytoplankton (LP); and
three zooplankton state variables – small (SZ), large (LZ) and predatory zooplankton (PZ).



Each biological state variable in NEMURO is an aggregated representation of taxonomically
diverse plankton groups that function similarly in the ecosystem. The phytoplankton community
in NEMURO is modeled as two functional types of obligate autotrophs: small phytoplankton (SP,
predominantly cyanobacteria and picoeukaryotes in the GoM) and large phytoplankton (LP,
diatoms). Small zooplankton (SZ) represent heterotrophic protists. Metazoan zooplankton are
divided into suspension-feeding mesozooplankton (LZ) and predatory zooplankton (PZ), which
also feed on LP and SZ. Here we assume that LZ and PZ are non-migratory. Heterotrophic bacteria
are implicitly represented in NEMURO by temperature-dependent decomposition rates, which
represent nitrification and remineralization. Sinking in NEMURO is restricted to PON and Opal
pools, and benthic processes are not included. Here, because of the large shelf area in the GoM,
we implemented a simple diagenesis of PON/OP to $NO_3/SiO_4$ and removal of PON/OP through
sedimentation, where 1% of the flux sinking out of bottom cell was removed and 10% converted
back into $NO_3/SiO_4$. However, we found that this had no significant impact on the model.
NEMURO uses nitrogen as a model "currency" since it is the major limiting macronutrient in
much of the ocean. Silica is also included as a potentially co-limiting nutrient for diatoms. For
more details on the specific processes represented and the interactions between state variables in
NEMURO, we direct readers to Kishi et al. (2007). All model equations are provided in the
Supplement to this manuscript.
NEMURO was chosen for the present study because it distinguishes SZ, LZ, and PZ, permitting a
detailed analysis of dynamics within the GoM zooplankton community and allowing for
investigation of multiple zooplankton functional types. In initial GoM simulations, we found that
default NEMURO parameterizations for the North Pacific (Kishi et al., 2007) substantially
overestimated both surface Chl and mesozooplankton biomass relative to observations. To a first
order, we attribute these differences to: 1) substantially higher temperatures in the GoM compared
with the North Pacific, which significantly increase decomposition and growth rates in the model
resulting in higher nutrient recycling and sustained elevated standing stocks of phytoplankton and
zooplankton near the surface, and 2) distinct differences in taxonomic composition of the
phytoplankton and zooplankton communities between the GoM and North Pacific with significant
differences in key parameter values associated with growth and grazing. Justification for each
parameter modification and steps of the model tuning process are outlined in Supplement **S2**, with





a summary of parameter values in **Table S2**. Biogeochemical model forcing, initial, and open
boundary conditions are also outlined in Supplement **S1**.

### 2.1.2   Modifications to the original biogeochemical model

To improve realism for application to the GoM, a total of five structural changes were made to the
original NEMURO transfer functions. First, we removed the SP to LZ grazing pathway. The
original SP state variable for the North Pacific represents nanophytoplankton (e.g.
coccolithophores), which can be important prey of copepods and other mesozooplankton. In the
GoM, however, cyanobacteria and picoeukaryotes (too small for direct feeding by most
mesozooplankton) comprise much of the phytoplankton biomass and hence are represented as SP
in our model. In addition to adding realism, this change in direct trophic connection between SP
and LZ allowed the model to produce a more realistic LP dominated phytoplankton community on
the shelf (see Discussion).
Next, quadratic mortality was replaced with linear mortality for all biological state variables with
the exception of predatory zooplankton (PZ). In biogeochemical models, quadratic mortality is
often used for numerical stability and/or to represent implicit loss terms to an un-modeled parasite
or predator that may covary in abundance with its prey (e.g. viral lysis of phytoplankton or
predation by un-modeled higher predators). However, grazing mortality is explicitly modeled in
NEMURO and viral mortality is generally not a substantial loss term for bulk phytoplankton
(Brum et al., 2014; Staniewski and Short, 2018). Quadratic mortality was retained for PZ, to
account for predation pressure of un-modeled planktivorous fish. We found that removal of
quadratic mortality for all other biological state variables led to more realistic mesozooplankton
biomass in the oligotrophic region (see Discussion).
The default ammonium inhibition term and light limitation functional form was replaced with a
more widely adopted parameterization. The exponential ammonium inhibition term in the nitrate
limitation function was replaced with the term described by Parker (1993), as has been done in
previous PBM studies (Fennel et al., 2006) due to the non-monotonic behavior of the default
NEMURO ammonium inhibition term. The default light limitation functional form was replaced
with the Platt et al. (1980) functional form that explicitly parameterizes photoinhibiton. This
formulation is implemented in newer versions of NEMURO, such as the code used in the Regional



Ocean Modeling System (ROMS) NEMURO biogeochemical package. Finally, to account for
photoacclimation and more accurately simulate Deep Chlorophyll Maximum (DCM) dynamics,
we replaced the constant C:Chl parameter with a variable C:Chl module where ratios for SP and
LP were allowed to vary based on the formulation described by Li et al. (2010), which considers
both light and nutrient limitation (see Supplemental). Herein, "default" NEMURO includes the
modified ammonium inhibition, light formulation and variable C:Chl model.
In total NEMURO has 75 parameters, 25 of which were modified in the present study. To tune
these parameters, we evaluated the model based on three observational benchmarks: surface Chl
estimated from seaWIFS, depth averaged mesozooplankton biomass from the SEAMAP dataset,
and DCM depth from the SEAMAP dataset. Chl and mesozooplankton biomass were chosen to
evaluate basin scale variability in plankton biomass while the DCM depth was chosen to evaluate
the vertical structure of the simulated ecosystem. We also considered expected patterns of size
structured phytoplankton community composition (i.e. SP:LP ratio), relative magnitudes of total
zooplankton grazing contributions, and the magnitude of loss terms for phytoplankton (grazing,
mortality, respiration, and excretion). Initial model tuning was carried out in an idealized one-
dimensional model before being implemented into the PBM. We outline each parameter change,
justification, and the resulting impact on the ecosystem benchmarks simulated by the one-
dimensional model in Supplement **Table S1**. Where possible, we modified parameters in groups
so that relative changes were consistent throughout the model (e.g. doubling all zooplankton
mortality terms). We also conducted a parameter sensitivity analysis to identify impacts of
parameter changes on the final three-dimensional PBM solution (herein referred to as NEMURO-
GoM) (Section 2.6).
**2.1.3 Description of the offline numerical environment**
To run large numbers of three-dimensional simulations efficiently for basin scale tuning, the
NEMURO-GoM was run offline using the MITgcm offline tracer advection package, which was
selected for this study as it has convenient packages for running offline simulations (McKinley et
al., 2004). That is, the dynamical equations of motion are not computed during the NEMURO-
GoM integration, but rather the physical prognostic variables (i.e., temperature, salinity, and three-
dimensional velocity fields) are prescribed from daily-averaged flow fields saved from a previous
hydrodynamic model integration. This allows the recycled use of flow fields leaving only the tracer



equations to be computed. In the offline MITgcm package, the prognostic variables provide input
to an advection scheme and mixing routine that conservatively handles offline advection and
diffusion of the biogeochemical tracer fields. MITgcm has many options for linear and non-linear
advection schemes. Here we use a $3^{rd}$ order direct space time flux limiting scheme. Sub grid-scale
mixing of the biogeochemical fields is handled offline through the nonlocal K-Profile
Parameterization (KPP) package based on mixing schemes developed by Large et al. (1994). For
more information about the MITgcm packages, we direct readers to the MITgcm manual
(http://mitgcm.org/).
Advantages of running PBMs in an offline environment include: 1) the physical time step in an
offline environment is no longer bound by the dynamical Courant–Friedrichs–Lewy numerical
stability criterion, allowing for longer time steps and fewer iterations; and 2) momentum equations
are not computed during the integration. Instead, the stability of the tracer advection scheme and
time scales needed to resolve biological/physical processes of interest set the limits on the time
steps and prescription frequencies of flow fields. When the physical time step is shorter than the
flow field prescription frequency, a simple linear interpolation of the flow fields is performed
inside the PBM between time steps. It is important to note that offline simulations of tracer
advection have been found to closely resemble online runs (that is, computed together with the
integration of the hydrodynamic model's prognostic equations) when the three-dimensional flow
fields are prescribed at a frequency that is at or below the inertial period for a region (Hill et al.,
266   2005).

In the present study, the NEMURO-GoM time step (30 minutes) is an order of magnitude greater
than the hydrodynamic model's (H-GoM, described in Section 2.1.4) baroclinic time step (120
seconds). For reference, the 20-year H-GoM simulation that supplied flow fields for the offline
NEMURO-GoM took a total of ~76 days to run to completion on 64 parallel cores. These time
requirements would increase considerably with the 11 additional biogeochemical tracers used in
NEMURO. In contrast, NEMURO-GoM including the 11 added tracers, ran significantly faster,
taking a total of ~50 h on 80 parallel cores. While computationally advantageous, it is important
to note that offline simulations inherently have greater input and output (I/O) demands that can
become bottlenecks in some applications.



### 2.1.4 Description of the ocean dynamical fields


The NEMURO-GoM is "forced" by daily averaged three-dimensional velocity, temperature, and
salinity fields from a preexisting 20-year (1993-2012) HYCOM (HYbrid Coordinate Ocean
Model) (Chassignet et al., 2003) regional GoM hindcast (H-GoM). H-GoM is based on version
2.2.99B of the HYCOM code, originally provided by the Naval Oceanographic Office
(NAVOCEANO) Major Shared Resource Center. H-GoM was run at $1/25^{th}$ (~4 km) degree
horizontal resolution with 36 vertical hybrid coordinate layers and assimilated historic, in situ, and
satellite observations. The domain encompasses the entire GoM and extends south of the Mexican-
Cuba Yucatan channel to 18 °N and as far east as 77 °W (**Fig. 1**). Further details on H-GoM
(experiment ID: GOMu0.04/expt_50.1) including details on model forcing and the main model
configuration file (i.e. blkdat.input_501) can be found at https://www.hycom.org.
The H-GoM flow fields were mapped from the HYCOM native vertical coordinate to z-levels used
by the MITgcm. The NEMURO-GoM was configured for 29 vertical z-levels in MITgcm (10-m
intervals from 0-150 m, 25-m intervals from 150-300 m, 50-m intervals from 300-500m, and 1000
m, 2000 m, ~4000 m). Mapping is performed by computing total zonal and meridional transports
across the lateral boundaries of each MITgcm grid cell (e.g., 0-10 m bin; which may include
multiple HYCOM layers) and then dividing by the area of the respective cell face. This vertical
mapping approach is consistent as both HYCOM and MITgcm use an Arakawa C-grid orientation
for model variables. The H-GoM bathymetry was adjusted such that no partial cells existed in the
domain to avoid thin cells. The continuity equation was subsequently used to calculate vertical
velocities. The use of transports in this approach ensures conservation and approximately identical
profiles of vertical velocity to those in H-GoM fields. For mapping of temperature and salinity
fields (used in the KPP mixing routine and for scaling biological temperature dependent rates) a
simple linear interpolation was performed.

### 2.2 Model validation


### 2.2.1 SeaWIFS observations used for model validation


A benchmark for surface Chl was determined using the Sea-Viewing Wide Field-of-View Sensor
(SeaWIFS) product from the Ocean Biology Processing Group (OBPG) of the National
Aeronautics and Space Administration (NASA). The product used here is the mapped, level-3,
daily, 9-km resolution product from 4 September 1997 to 10 December 2010 processed according
to the algorithm of Hu et al. (2012). To compute model-data point-to-point comparisons, we take
the corresponding daily averaged simulated surface Chl field and interpolate to the SeaWIFS grid
before applying the daily cloud coverage mask corresponding to the matching SeaWIFS image. In
total 4,291 daily images consisting of 22,244,513 non-zero Chl cell values (herein referred to
seaWIFS measurements) were used to validate the PBM. Approximately 500-1200 daily model-
data point-to-point comparisons were made for each SeaWIFS grid cell.

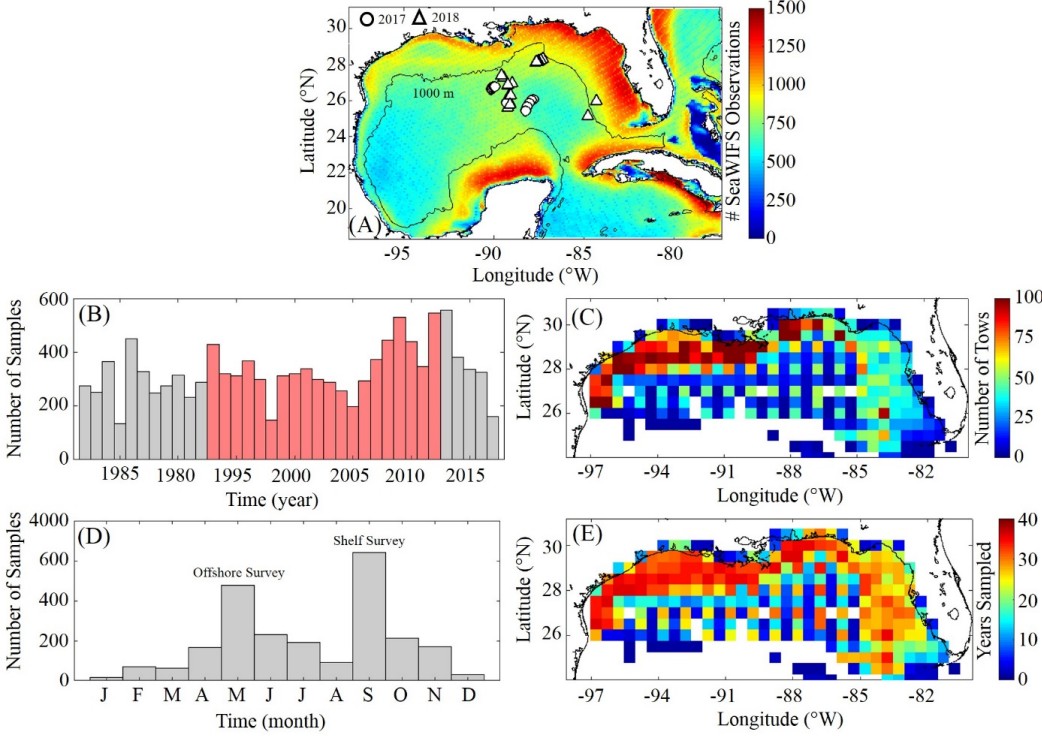


**Figure 1 (A-E):** Spatial and temporal coverage of all observational data sets used for model
validation. Total number of non-zero SeaWIFS values from the level 3 product from 4 September
1997 to 10 December, 2010 along with cruise sample locations collected during May, 2017
(circles) and 2018 (triangles) (A). Total annual sampling of the SEAMAP surveys from 1983-2017
(B) with samples overlapping with the PBM simulation period denoted in red. Total sample density
within each 0.5° x 0.5° box (C). Total seasonal sampling (D). Number of years with at least one
sample (E). 1000 m isobaths and coastline are denoted by black continuous lines.



### 2.2.2 SEAMAP observations used model validation

To evaluate model mesozooplankton biomass estimates, we used zooplankton biomass data
collected during SEAMAP surveys in the northern and central GoM. In total, 11,781 zooplankton
tows were collected from 1983-2017 with two main annual surveys consisting of a spring offshore
and fall shelf sampling grid (**Fig. 1**). These samples were used to generate a climatology which
was used to compare with simulated mesozooplankton climatology. On average, SEAMAP
surveys collected approximately 300 samples per year with a specific sampling array in the
offshore surveys and more general spatial sampling coverage on the shelf. Of these samples, 6,835
were used for direct point-to-point model-data comparisons. Zooplankton biomass samples were
collected using standard gear consisting of a 61 cm diameter bongo frame fitted with two 333 μm
mesh nets. This gear is fished in a double-oblique tow pattern from the surface down to 200 m or
5 m off the bottom and back to the surface. During 82 tows in nearshore and oligotrophic regions,
additional samples were collected using a 202 μm mesh net concurrently with the standard 333 μm
mesh net. Of these samples roughly half were collected in the oligotrophic GoM. The average ratio
between 333 and 202 samples ($0.5093 \pm 0.12$) was used to convert biomass measurements from
the 333 μm mesh samples so that direct comparisons could be made with simulated
mesozooplankton biomass estimates. In this study we consider SZ size to be < 200 μm, LZ size to
be 0.2-1 mm, and PZ size to be 1-5 mm. Zooplankton biomasses from SEAMAP surveys were
originally quantified as displacement volumes (DV). Carbon mass (CM) equivalents were
subsequently calculated as $\log_{10}(CM) = (\log_{10}(DV) +1.434)/0.820$ (Wiebe, 1988; Moriarty and
O'Brien, 2013). CM estimates were converted to model units (mmol N m$^{-3}$) assuming Redfield
C:N ratio. Simulated mesozooplankton model fields were similarly depth integrated to the bottom
or 200 m to generate the model mesozooplankton biomass climatology or to the sample depth
when performing point-to-point comparisons.
Vertical depth profiles of Chl were also approximated at standard stations during SEAMAP
surveys using a SeaBird WETStar pumped fluorometer attached to a CTD. These profiles were
used to determine the depths of the fluorescence maxima, which were then compared directly to
simulated DCM depths at corresponding times and locations. In total, 2,435 profiles were taken
from 2003-2012, with 1,052 profiles overlying bottom depths >1000 m. Profiles were available
for earlier SEAMAP surveys; however, no standard QA/QC protocol for fluorometer data was in



place prior to 2003. Model-data agreement for DCM magnitude could not be investigated as the
fluorometer was not calibrated before each cruise.

### 2.2.3    Process rate measurements used for model validation

Although in situ rate measurements are made much less frequently than biological standing stock
measurements, they offer very powerful constraints for validating the internal dynamics of a
biogeochemical model (Franks, 2009).  Consequently, we made phytoplankton and zooplankton
rate measurements on two cruises in the open ocean GoM in May 2017 and 2018 and used these
measurements to validate the model (**Fig. 1A**). Since the cruise sampling does not overlap with
our NEMURO-GoM simulation period, we sampled the model at corresponding locations and
times of the year for all 20 years of the simulation to investigate model-data comparisons. On these
cruises, we utilized a quasi-Lagrangian sampling scheme to investigate plankton dynamics in the
oligotrophic GoM. Two drifting arrays (one sediment trap array and one in situ incubation array)
were then deployed to serve as a moving frame of reference during ~4-day studies ("cycles")
characterizing the water parcel (Landry et al., 2009; Stukel et al., 2015).  During these cycles, we
measured daily profiles of Chl, photosynthetically active radiation, phytoplankton growth rates
and productivity, protistan grazing rates, and size-fractionated mesozooplankton biomass and
grazing rates.
Protistan grazing rates were measured using the two-point, "mini-dilution" variant of the
microzooplankton grazing dilution method (Landry et al., 1984, 2008; Landry and Hassett, 1982).
Briefly, one 2.8-L polycarbonate bottle was gently filled with whole seawater taken from six
depths (from the surface to the depth of the mixed layer).  A second 2.8-L bottle was then filled
with 33% whole seawater and 67% 0.2-μm filtered seawater. Both bottles were then placed in
mesh bags and incubated in situ at natural depths for 24 h.  These experiments were conducted on
each day of the ~4-day cycle.  After 24 h, the bottles were retrieved, filtered onto glass fiber filters,
and Chl concentrations were determined using the acidification method (Strickland and Parsons.,
1972).  Net growth rates ($k=\ln(Chl_{final}/Chl_{init})$) in each bottle were then determined relative to initial
Chl samples.  Phytoplankton specific mortality rates resulting from the grazing pressure of protists
were calculated as $m = (k_d - k_0)/(1-0.33)$, where $k_d$ is the growth rate in the dilute bottle and $k_0$ is
the growth rate in the control bottle. Phytoplankton specific growth rates were calculated as $\mu = k_0$
$+ m$. For additional details, see Landry et al. (2016) and Selph et al. (2016). Phytoplankton net



381 primary production was quantified at the same depths by $H^{13}CO_3^-$ uptake experiments. Triplicate

382 2.8-L polycarbonate bottles and a fourth "dark" bottle were spiked with $H^{13}CO_3^-$ and incubated in

383 situ for 24 h at the same sampling depths as for the dilution experiments. Samples were then

384 filtered, and the $^{13}C:^{12}C$ ratios of particulate matter were determined by isotope ratio mass

385 spectrometry.

386 Size-fractionated mesozooplankton biomass and grazing rates were determined from daily day-

387 night paired oblique ring-net tows (1-m diameter, 202-µm mesh) to a depth of 110 m. Upon

388 recovery, the sample was anesthetized using carbonated water, split using a Folsom splitter,

389 filtered through a series of nested sieves (5, 2, 1, 0.5, and 0.2 mm), filtered onto preweighed 200-

390 µm Nitex filters, rinsed with isotonic ammonium formate to remove sea salt, and flash frozen in

391 liquid nitrogen. In the lab, defrosted samples were weighed for total wet weight, and subsampled

392 in duplicate (wet weight removed) for gut fluorescence analyses. The remaining wet sample was

393 dried and subsequently reweighed and combusted for CHN analyses to determine total dry weight

394 and C and N biomasses. Gut fluorescence subsamples were homogenized using a sonicating tip,

395 extracted in acetone, and measured for Chl and phaeopigments using the acidification method.

396 The phaeopigment concentrations in the zooplankton guts were the basis for calculated grazing

397 rates using gut turnover times based on temperature relationships for mixed zooplankton

398 assemblages. For additional details, see Décima et al. (2011) and Decima et al. (2016).

399 **2.3 Description of the parameter sensitivity experiments**

400 After validating the PBM, a parameter sensitivity analysis consisting of 18 numerical experiments

401 was conducted to evaluate how robust the final model solution was to parameter changes. For

402 each experiment, the PBM was configured to simulate four years starting in January 2002. This

403 time period was concurrent with SeaWIFS and SEAMAP sample coverage. Parameter sensitivity

404 experiments were initialized from our standard NEMURO-GoM run at 1 January 2002. The PBM

405 with each parameter change(s) was then allowed to spin up for one year. The last three years (i.e.

406 2003-2005) were subsequently used for the parameter sensitivity analysis. Direct point-to-point

407 comparisons were made between model estimates and observations at corresponding sample times

408 and locations during the model integration. In total, 4,646,459 SeaWIFS Chl measurements, 741

409 SEAMAP mesozooplankton tows, and 481 SEAMAP fluorescence profiles were used to evaluate

410 model sensitivity. To better capture relative differences between model and observations across





coastal and oligotrophic GoM regions, a $\log_{10}$ transformation was applied to Chl and
mesozooplankton biomass model-data comparisons before calculating Taylor and Target diagram
statistics. Point-to-point model-data comparisons were also made using the 20-year PBM output,
which included all available data (i.e. 22,244,513 SeaWIFS Chl measurements, 6,835 SEAMAP
mesozooplankton tows, and 2,435 SEAMAP fluorescence profiles). Configurations for each
parameter sensitivity experiment are outlined in **Table S3**.
**3.0      Results**
**3.1      Regional phytoplankton biomass model-data comparisons**
Model surface Chl estimates demonstrate strong agreement with satellite observations (**Fig. 2**).
Spatial covariance between SeaWIFS climatology and model surface Chl climatology (calculated
with daily cloud cover mask applied) is found to be statistically significant ($p < 0.01$) with a
correlation ($\rho$) of 0.72. When model estimates are compared to all 22,244,513 SeaWIFS
measurements at corresponding times and locations (i.e. daily grid cell pairs), we find a $\rho$ value of
0.50 ($p < 0.01$). To facilitate more detailed model-data comparisons, the GoM domain was divided
into an oligotrophic region ($\geq$1000 m bottom depth) and a shelf region (<1000 m bottom depth).
In the oligotrophic region, the correlation between model-data daily grid cell pairs is significant
but weak ($\rho = 0.17$, $p < 0.01$) as a result of relatively low large-scale spatial variability, and hence
dominance at the mesoscale.  However, bias is quite low (-0.014 mg Chl m$^{-3}$) equivalent to 10%
of the observed mean. In the shelf region, the correlation is higher ($\rho = 0.47$, $p < 0.01$) yet the bias
is greater (+0.90 mg Chl m$^{-3}$) equivalent to 92% of the mean. Previous GoM studies have
determined $\rho$ values based on monthly averages and for reference we calculate them here. Based
on 30-day averages we find a $\rho$ value of 0.70 ($p < 0.01$) for the oligotrophic region and 0.26 ($p <$
0.01) for the shelf region.
In addition to resolving the dominant spatiotemporal variability, the model also captures the
amplitude of the seasonal surface Chl signal reasonably well. In the oligotrophic region, the model
accurately estimates the observed annual surface Chl minimum (Model: 0.065 $\pm$ 0.005 vs.
SeaWIFS: 0.065 $\pm$ 0.007 mg Chl m$^{-3}$) while slightly underestimating the observed annual
maximum (Model: 0.47 $\pm$ 0.15 vs. SeaWIFS: 0.75 $\pm$ 0.55 mg Chl m$^{-3}$). When model estimates for
the entire oligotrophic region are taken into account (i.e. not restricted to satellite measurement
locations and times), we find the annual minimum develops in early September while the annual



maximum develops in late January (**Table 1**). In the shelf region, greater model-data mismatch
exists for surface Chl where the model overestimates the observed annual minimum by 15%
(Model: $0.23 \pm 0.09$ vs. SeaWIFS: $0.20 \pm 0.07$ mg Chl m$^{-3}$) and the observed annual maximum by
102% (Model: $8.09 \pm 1.31$ vs. SeaWIFS: $4.01 \pm 1.23$ mg Chl m$^{-3}$). Here, we find the annual surface
Chl seasonal cycle is almost completely out of phase with the oligotrophic region with the annual
minimum developing during early February and the annual maximum developing at the end of
July (**Table 1**).

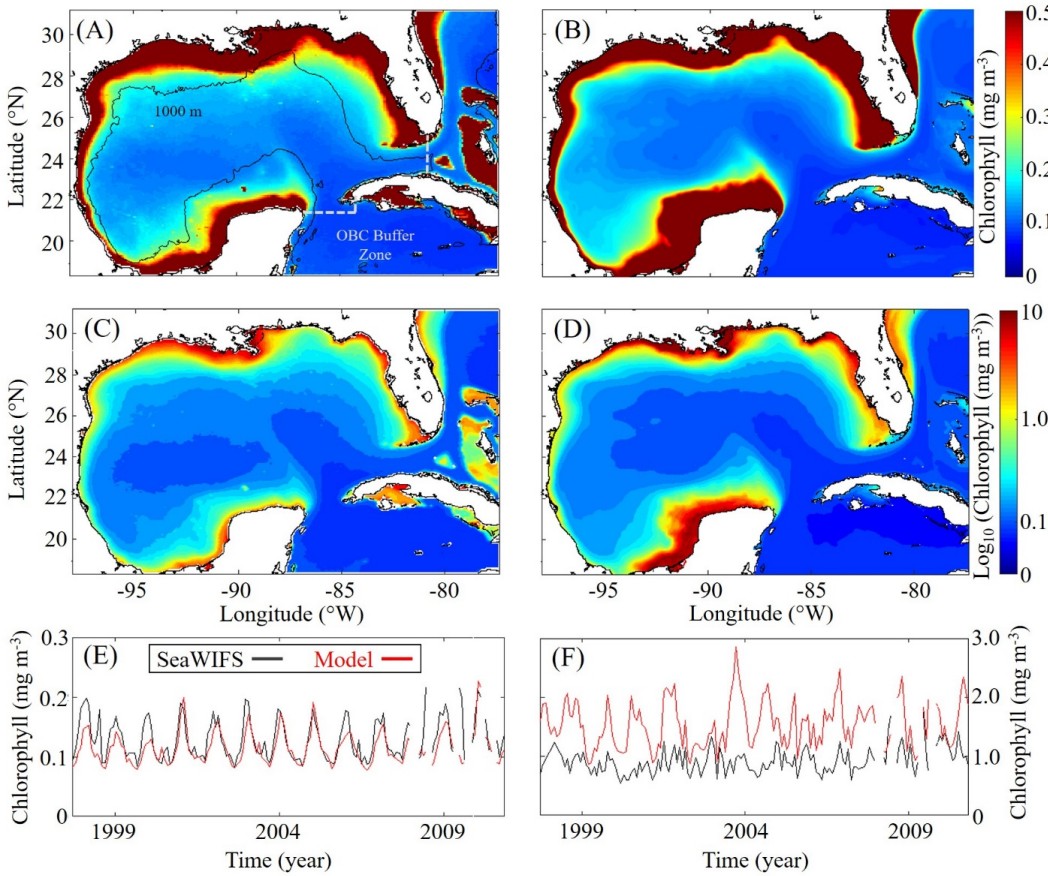


**Figure 2 (A-F):** Comparison of surface chlorophyll (mg m$^{-3}$) between SeaWIFS observations and
model from 4 September 1997 to 10 December 2010. Average SeaWIFS chlorophyll (A). Average
model estimated surface chlorophyll (B). Log$_{10}$ of the average SeaWIFS chlorophyll (C). Log$_{10}$ of
the average model estimated surface chlorophyll (D). Time series of simulated 30-day average



surface chlorophyll (red) and SeaWIFS observations (black) for bottom depths ≥1000 m (E) and
bottom depths <1000 m (F). The 1000 m isobaths and coastline are denoted by black lines.
**Table1:** Average seasonal minimum and maximum values in the model (1993-2012) and the day
of year in which they occur for surface chlorophyll (mg m$^{-3}$), depth integrated phytoplankton
biomass (mg C m$^{-2}$), depth integrated net primary production (mg C m$^{-2}$ d$^{-1}$), depth integrated
mesozooplankton biomass (mg C m$^{-2}$), and depth integrated mesozooplankton secondary
production (mg C m$^{-2}$ d$^{-1}$) calculated by spatially averaging daily fields over the oligotrophic
region (upper half of table) and shelf region (lower half of table). Day of year values are in the
format "day/month ± days."

| | Daily Field Value | | Day of Year | |
|---|---|---|---|---|
| Diagnostic (Oligotrophic) | Annual Min. | Annual Max. | Day of Min. | Day of Max. |
| Surface Chlorophyll | 0.09 ± 0.005 | 0.27 ± 0.06 | 9/9 ± 23 | 1/29 ± 13 |
| Phytoplankton Biomass | 2300 ± 130 | 3600 ± 140 | 12/26 ± 7 | 4/29 ± 17 |
| Net Primary Production | 290 ± 70 | 1000 ± 120 | 12/31 ± 12 | 7/6 ± 27 |
| Mesozooplankton Biomass | 1000 ± 40 | 1400 ± 90 | 1/1 ± 4 | 5/19 ± 18 |
| Secondary Production | 18 ± 4 | 68 ± 10 | 12/31 ± 10 | 6/4 ± 15 |
| Diagnostic (Shelf) | Annual Min. | Annual Max. | Day of Min. | Day of Max. |
| Surface Chlorophyll | 1.96 ± 0.15 | 3.00 ± 0.30 | 2/8 ± 37 | 7/31 ± 58 |
| Phytoplankton Biomass | 3200 ± 290 | 5200 ± 440 | 1/1 ± 9 | 7/18 ± 11 |
| Net Primary Production | 750 ± 120 | 2000 ± 220 | 12/31 ± 8 | 7/21 ± 14 |
| Mesozooplankton Biomass | 670 ± 70 | 1100 ± 90 | 12/29 ± 7 | 5/23 ± 25 |
| Secondary Production | 94 ± 17 | 270 ± 28 | 12/31 ± 6 | 7/20 ± 16 |


The model also captures the vertical variability in phytoplankton biomass reasonably well, falling
within one standard deviation of the observed data. When model estimates of DCM depth are
compared to all 2,435 SEAMAP CTD casts at corresponding sample times and locations, we find
a statistically significant correlation ($\rho = 0.59$, $p < 0.01$) with the observed maximum fluorescence
depth. The observed DCM depth ranged from the surface to 143 m while model values show a
similar variability ranging from the surface to 163 m.  In the oligotrophic region, we find the model
overestimates the DCM (Model: 95 ± 20 m vs. SEAMAP: 80 ± 25 m) and has a $\rho$ value of 0.38 (p
< 0.01) with a bias of 15 m equivalent to 19% of the observed mean. In the shelf region, the model



also overestimates DCM depth (Model: $63 \pm 26$ m vs. SEAMAP: $53 \pm 23$ m) and has a ρ value of
0.49 ($p < 0.01$) with a bias of 10 m equivalent to 19% of the observed mean.
**3.2      Regional zooplankton biomass model-data comparisons**

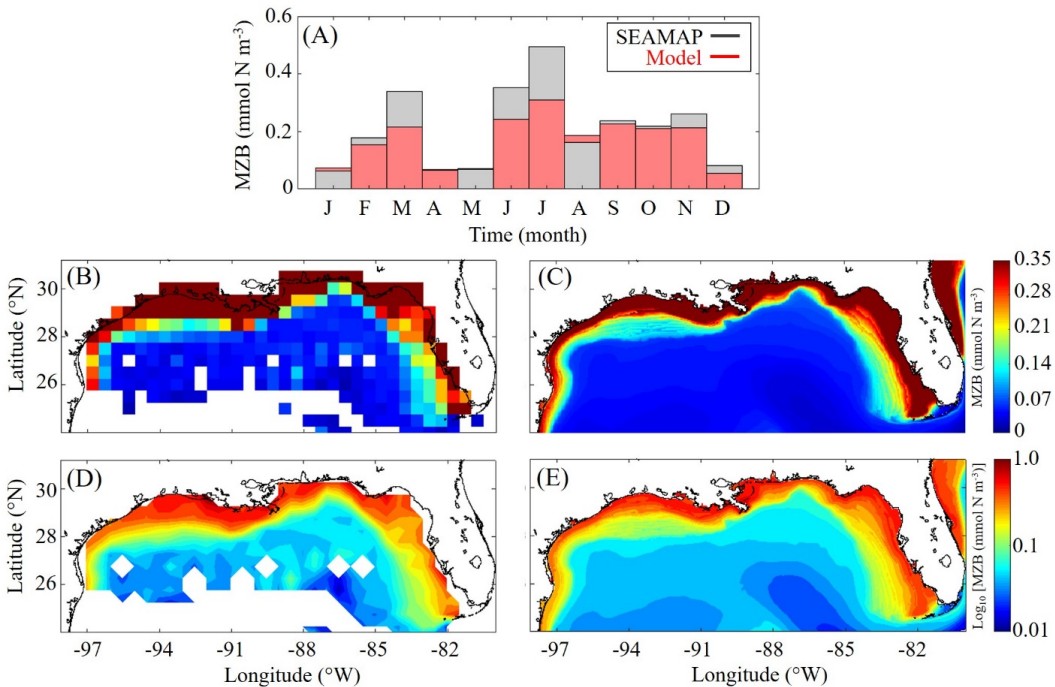


**Figure 3 (A-E):** Comparison of climatological depth-averaged mesozooplankton biomass (MZB,
mmol N m$^{-3}$) between SEAMAP observations (left) and model output (right). Monthly average
MZB samples organized by month (A). Monthly variability is not representative of seasonality as
sampling locations change between months. MZB from all SEAMAP tows (B). MZB 20-year
model average (C). Log$_{10}$ of SEAMAP MZB (D). Log$_{10}$ of model MZB (E).
Model mesozooplankton biomass (i.e. LZ + PZ) fields compare well with observations in both the
oligotrophic and shelf region (**Fig. 3**). Spatial covariance between SEAMAP climatology and
model climatology of depth-averaged mesozooplankton biomass is statistically significant ($p <$
0.01) with a ρ value of 0.90. When model estimates were compared to SEAMAP tows at
corresponding sample times and locations for the 6,835 measurements overlapping with the
simulation period, the ρ value is 0.55 ($p < 0.01$). In the oligotrophic region, the model slightly



overestimates mesozooplankton biomass (Model: $4.09 \pm 1.82$ mg C m$^{-3}$ vs. SEAMAP: $3.52 \pm 3.44$
mg C m$^{-3}$) with $\rho$ value of 0.23 (p < 0.01) and bias of 0.57 mg C m$^{-3}$ equivalent to 16% of the
observed mean. Conversely, in the shelf region the model underestimates mesozooplankton
biomass (Model: $17.40 \pm 13.58$ mg C m$^{-3}$ vs. SEAMAP: $20.91 \pm 24.62$ mg C m$^{-3}$), with a $\rho$ value
of 0.49 (p < 0.01) and a bias of -3.5 mg C m$^{-3}$ equivalent to 17% of the observed mean. We note
that model estimates and SEAMAP measurements also compare well with mesozooplankton
biomass measurements (0.2-5 mm) obtained in the oligotrophic region from independent May,
2017 and 2018 cruises (Model: $5.55 \pm 2.87$ mg C m$^{-3}$ vs. Cruise: $4.33 \pm 2.28$ mg C m$^{-3}$).
Although seasonal cycles in the oligotrophic and shelf regions could not be derived from the
SEAMAP dataset given the significant differences in sampling locations over the course of a year,
we investigated model-data mismatches for each month. We find the model closely matches or
slightly underestimates depth-averaged mesozooplankton biomass throughout most of the year,
with the exception of January, May, and August (**Fig. 3A**). The greatest model-data mismatch
occurs during the months of March, June, July, and December, where the model underestimates
depth-averaged mesozooplankton biomass by approximately 35%. Unlike phytoplankton biomass,
the total mesozooplankton biomass (i.e. depth-integrated) seasonality is similar in both regions of
the GoM. In the oligotrophic region, the annual mesozooplankton biomass minimum (maximum)
develops at the beginning of January (middle of May) while in the shelf region, the annual
minimum (maximum) develops in late December (near the end of May) (**Table 1**).

### 3.3      Phytoplankton growth and zooplankton grazing model-data comparisons

To further constrain the phytoplankton and zooplankton community simulated by the PBM, we
utilized in situ measurements of the planktonic community during Lagrangian process studies
conducted on two cruises in the oligotrophic GoM during May 2017 and 2018. First, we compared
the relative proportions of LZ and PZ biomass to four discrete size classes measured at sea (**Fig.**
**4A, C**). In total, 40 oblique bongo net tows (16 in 2017 and 24 in 2018) sampled the oligotrophic
GoM mesozooplankton community from near surface to a depth ranging from 100 - 135 m. When
the model is sampled yearly corresponding to cruise measurement locations and day of the year,
we find nearly identical size distributions when assuming that LZ approximates the smallest two
size classes of mesozooplankton sampled ("small mesozooplankton", 0.2-1.0-mm) and PZ
approximates the largest two size classes ("large mesozooplankton", 1.0-5.0 mm). In both



observations and model estimates approximately 40% and 60% of the mesozooplankton
community is composed of LZ and PZ, respectively. In the field data, small mesozooplankton
biomass varied from 33 to 46 % (median = 40%, at 95% C.I.), while model estimates of LZ
biomass vary from 31 to 46% (median = 40%). Large mesozooplankton biomass in the field data
varied from 54 to 67% (median = 60%), while model estimates of PZ biomass vary from 54 to
69% (median = 60%).

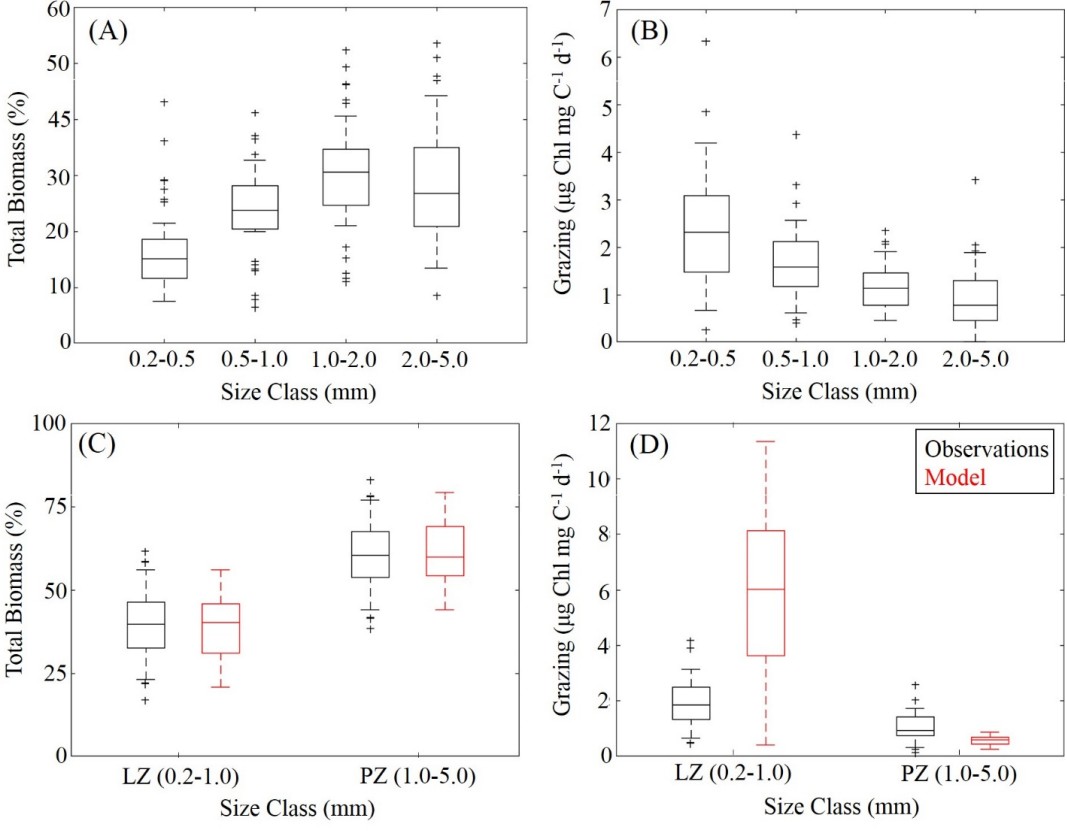


**Figure 4 (A-D):** A summary of field (black) and model (red) estimates of mesozooplankton size-
fractioned biomass and grazing rates. Mesozooplankton size-fractioned biomass as a percent of
total biomass for each of the four size classes measured at sea in May, 2017 and 2018 (A).
Corresponding mesozooplankton specific grazing rates for each of the four size classes (B). Field
data aggregated into two size classes for direct comparison with model biomass estimates for large
(LZ) and predatory (PZ) mesozooplankton (C). Similarly, model data comparison of specific





grazing rates by large and predatory zooplankton to aggregated field estimates (D). Whiskers
extend to 95% confidence interval. Outliers for model estimates are not shown.
We also measured the specific grazing rates of each size class using the gut pigment approach.
Field measurements showed that specific grazing rates consistently decreased with increasing
mesozooplankton size-class (**Fig. 4B**). To compare specific grazing rates in the model to field
measurements ($\mu$g Chl mg C$^{-1}$ d$^{-1}$), we computed grazing on LP by LZ and PZ at each depth.
Grazing terms were converted into units of Chl using the model estimated C:Chl ratio for LP before
being depth-integrated to the corresponding net tow depth and normalized to simulated depth-
integrated LZ and PZ biomasses. We find that model mesozooplankton grazing estimates capture
the general trend of decreased specific grazing rates with increasing mesozooplankton size (**Fig.**
**4D**). However, the model overestimates grazing by small mesozooplankton while underestimating
grazing by large mesozooplankton. In the field data, small mesozooplankton grazing varied from
1.34 to 2.51 $\mu$g Chl mg C$^{-1}$ d$^{-1}$ (median = 1.85) while model estimates of LZ grazing rates vary
from 3.64 to 8.14 $\mu$g Chl mg C$^{-1}$ d$^{-1}$ (median = 6.01). Field measurements of large
mesozooplankton grazing varied from 0.76 to 1.44 $\mu$g Chl mg C$^{-1}$ d$^{-1}$ (median = 0.94), while model
estimates of PZ grazing vary from 0.44 to 0.70 $\mu$g Chl mg C$^{-1}$ d$^{-1}$ (median = 0.58). In terms of total
mesozooplankton grazing, average grazing in the field was found to be 1.38 $\pm$ 0.59 $\mu$g Chl mg C$^{-1}$
d$^{-1}$, while the model average is 2.99 $\pm$ 2.20 $\mu$g Chl mg C$^{-1}$ d$^{-1}$. This model-data mismatch likely
results from the fact that, as formulated in NEMURO, LZ and PZ do not necessarily reflect size
classes of mesozooplankton, but rather functional types. In reality, there is substantial overlap
between taxonomic groups with different functional roles and sizes (see Discussion).
In addition to measuring the mesozooplankton community, specific phytoplankton growth rates
and specific phytoplankton mortality due to microzooplankon grazing were measured at sea using
the microzooplankon grazing dilution method, and net primary production (NPP) was measured
with H$^{13}$CO$_3^-$ uptake experiments. We find the model underestimates phytoplankton growth and
microzooplankton grazing while overestimating NPP (**Fig. 5A, B**). This model-data mismatch may
be driven in part by model errors in simulated vertical patterns of phytoplankton growth rates. We
note that model results consistently predict enhanced growth rates at the DCM, while the field
measurements showed surface enhancement of growth rates or relatively constant growth rates
with depth. We believe the collocation of high growth rates at the DCM estimated by the model



may reveal a fundamental issue with how biogeochemical models simulated DCM dynamics. This
collocation could explain the lower specific growth rates despite higher NPP we find in the model
(see Discussion).

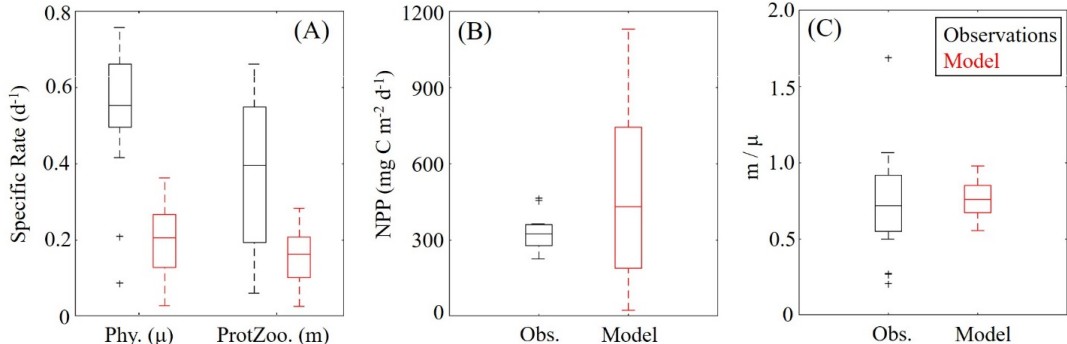


**Figure 5 (A-C):** Specific phytoplankton growth ($\mu$, d$^{-1}$) and microzooplankon grazing (m, d$^{-1}$)
between model (red) and field data (black) (A). Depth-integrated net primary production (mg C
m$^{-2}$ d$^{-1}$) (B). The fraction of phytoplankton growth that is grazed by protists in the model and field
data (C). Whiskers extend to the 95% confidence intervals. Outliers for model estimates are not
shown.
Phytoplankton specific growth rates in dilution experiments varied from 0.50 to 0.66 d$^{-1}$ (median
= 0.55 d$^{-1}$) while model estimates of phytoplankton (SP+LP) specific growth rates are lower and
vary from 0.13 to 0.27 d$^{-1}$ (median = 0.21 d$^{-1}$). In terms of microzooplankton grazing, field data
varied from 0.19 to 0.55 d$^{-1}$ (median = 0.39 d$^{-1}$) while model estimates of SZ grazing are also lower
and vary from 0.10 to 0.21 d$^{-1}$ (median = 0.16 d$^{-1}$). NPP estimates between model and data show
better agreement where field data varied from 275.61 to 360.09 mg C m$^{-2}$ d$^{-1}$ (median = 321.44 mg
C m$^{-2}$ d$^{-1}$) while model estimates vary from 189.75 to 741.04 mg C m$^{-2}$ d$^{-1}$ (median = 430.96 mg C
m$^{-2}$ d$^{-1}$). Although we find the model underestimates specific phytoplankton growth and
microzooplankton grazing rates, the relative proportion of NPP being consumed by protists
compares reasonably well to field measurements (**Fig. 5C**). The proportion of NPP grazed in field
data varied from 55% to 92% (median = 72%), while model estimates vary from 67% to 85%
(median = 76%). Notably, the model average proportion of phytoplankton production consumed
by protists closely matches the mean for all tropical waters reported by Calbet & Landry (2004).





When specific phytoplankton mortality due to mesozooplankton grazing was calculated at cruise
sample locations, we find that mesozooplankton grazing accounts for 13 ± 8 % which also closely
agrees with the global average (Calbet et al., 2001).
**3.4    Parameter sensitivity analysis**
To evaluate model sensitivity, we investigated the impact of parameter changes on model estimates
over the entire GoM domain and the oligotrophic region, specifically. The separate analysis of the
oligotrophic region was undertaken for two reasons: 1) this region is an area where low
mesozooplankton biomass likely leads to particularly strong prey limitation for fish, their larvae,
and other higher trophic levels and 2) the substantially higher biomass and variability on the shelf
dominates region-wide mean estimates. In comparison to default NEMURO, the NEMURO-GoM
produces estimates of surface Chl, depth averaged mesozooplankton biomass, and DCM depth that
more closely agree with observations (**Fig. 6**). During the parameter sensitivity experiments
SEAMAP observations in the oligotrophic region were almost always located near the Loop
Current which is strongly influenced by the southern open boundary condition. Hence, differences
between simulations were difficult to quantify. Additionally, since mesozooplankton biomass
observations is a depth averaged metric differences between simulations can appear small despite
extreme differences in the vertical distribution of biomass.
All parameter sensitivity experiment configurations are outlined in Supplement **Table S3.** Of the
18 sensitivity experiments, the greatest model overestimation of surface Chl occurs when default
α values (slope of the photosynthesis-irradiance curve) are included in NEMURO-GoM (**Fig. 6A-**
**D**). In default NEMURO, SP and LP α values are an order of magnitude lower (0.01). When default
α values are included in the NEMURO-GoM, they restrict the depth range where phytoplankton
can grow, resulting in substantially shallower DCM depths than observed. Subsequently, the
nitracline becomes unrealistically shallow (~25 m in the oligotrophic region), allowing nutrients
to mix readily into surface water and support higher phytoplankton biomass. The greatest model
underestimation of surface Chl occurs when default quadratic mortality is implemented in the
NEMURO-GoM. Although quadratic mortality tends to increase the lower limit of phytoplankton
biomass, it also increases zooplankton standing stocks which, in this case, allows zooplankton to
graze phytoplankton to unrealistically low levels. We find the exact opposite is true for
mesozooplankton biomass. The greatest overestimation of depth-averaged mesozooplankton



biomass occurs when default quadratic mortality is included in the NEMURO-GoM. Conversely,
when default α values are included we find the largest underestimation of mesozooplankton
biomass as a result of low phytoplankton biomass at depth (**Fig. 6E-H**).

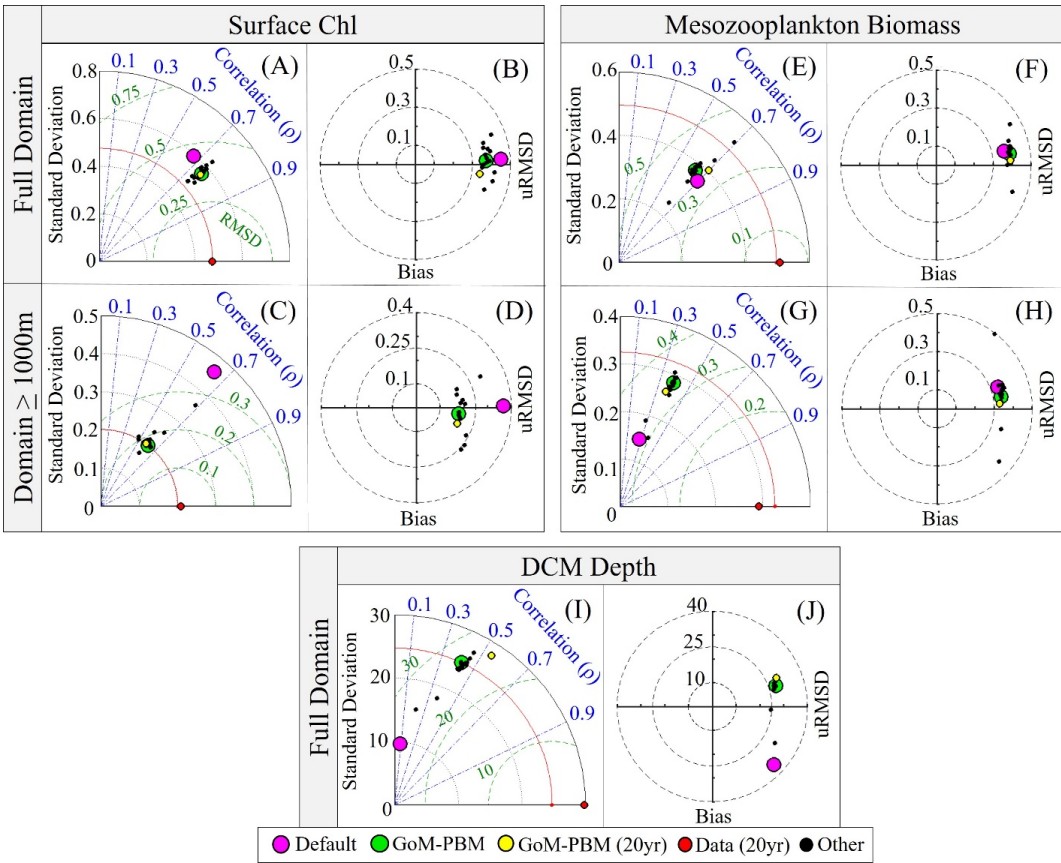


**Figure 6 (A-J):** Taylor and Target diagrams comparing 18 parameter sensitivity experiments
(black dots) against observations of surface Chl (top left, A-D) depth-averaged mesozooplankton
biomass (top right, E-H) and deep chlorophyll maximum depth (bottom center, I & J). Each panel
contains Taylor diagrams (left) and Target diagrams (right). The top two panels are further divided
based on analysis of all data (top) and with bottom depths $\geq$ 1000 m (bottom). The red arc in Taylor
diagrams signifies the standard deviation of all observations in the last three years of the four-year
parameter sensitivity experiments (2002-2006). A $\log_{10}$ transform is applied to surface chlorophyll
and depth-averaged mesozooplankton before computing model-data statistics.



We also investigated the influence of parameter changes on simulated DCM depth (**Fig. 6I, J**). For
this analysis, we did not isolate the oligotrophic region because average DCM depth does not vary
as substantially as biomass between the shelf and oligotrophic regions (i.e., the shelf does not
dominate the region-wide signal). In contrast to surface Chl and mesozooplankton biomass, default
mortality does not strongly influence DCM depth. However, when default α values are included,
the model substantially underestimates the actual DCM depth and the standard deviation of DCM
depth as expected. In the NEMURO-GoM, tuned values lead to substantial improvement in DCM
depth, with a standard deviation quite close to observations and a substantially improved ρ value
(**Fig. 6I**). However, the tuned parameter set results in a small positive bias in DCM depth (i.e.,
deeper than measured DCM by ~10 m), although this was less significant than the negative bias in
DCM depth of default NEMURO (i.e., shallower DCM than observations by ~25 m).
**3.5    Simulated mesozooplankton diet and secondary production**
Trophic level estimates provide a measure of the cumulative diet for mesozooplankton. We
estimated mesozooplankton trophic level in the model by computing the dietary contributions of
each prey in LZ (i.e. LP and SZ) and PZ diets (i.e. LP, SZ, and LZ) while assuming that the trophic
level of LP = 1 and SZ = 2. In the oligotrophic region, both LP and SZ contribute approximately
50% to LZ diet, as indicated by average LZ trophic level near 2.5 (2.54 ± 0.02) (**Fig. 7A**). In the
same region, PZ have a trophic level of 2.78 ± 0.04 indicating a higher contribution of zooplankton
to their diet (i.e. SZ and/or LZ) (**Fig. 7B**). In the shelf region, LZ are more herbivorous, as indicated
by a decrease in trophic level to 2.31 ± 0.01, while PZ are more carnivorous, as indicated by an
increase in trophic level to 2.90 ± 0.04.
Although there is little evidence in the annual average for LZ diets dominated by zooplankton
(trophic level ~3 as commonly found in PZ diets), we commonly find regions in instantaneous
fields during both winter and summer where SZ are the dominant prey source for LZ (**Fig. 7C, E**).
These regions, typically in the Loop Current or Loop Current Eddies (LCEs), highlight the episodic
importance of heterotrophic protists as prey sources for small mesozooplankton in the GoM. High
proportions of SZ in LZ diets can be attributed to the competitive advantage of SP over LP in
extremely low nutrient environments such as in the Loop Current. Instantaneous fields also reveal
that phytoplankton can be an important prey source for PZ as well. This is particularly the case





during summer, as indicated by trophic levels of around 2.5 in the western oligotrophic GoM (**Fig.**
**7F**).

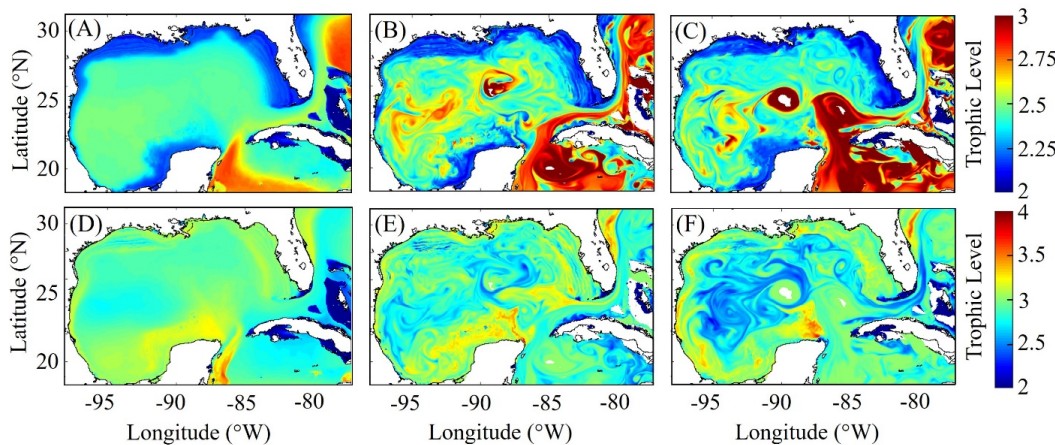


**Figure 7 (A-F):** Trophic levels of simulated large zooplankton (LZ, top) and predatory
zooplankton (PZ, bottom). Annual-average trophic positions of LZ (A) and PZ (D). Instantaneous
trophic positions of LZ (B) and PZ (E) for winter conditions on 4 February 2012. Instantaneous
trophic positions of LZ (C) and PZ (F) for summer conditions on 5 August 2011.
In addition to strong variability in trophic positions, there are also regions in the oligotrophic GoM,
most clearly in the centers of LCEs during summer, where the model predicts no feeding by
mesozooplankton (**Fig. 8E**). The convergent anti-cyclonic circulation of LCEs is typically
associated with low phytoplankton biomass, which at times may fall near or below feeding
thresholds in the NEMURO grazing formulation. This formulation is designed to simulate
suppression of feeding activity for zooplankton at mean prey densities that cannot support the
energy expended while searching for prey.

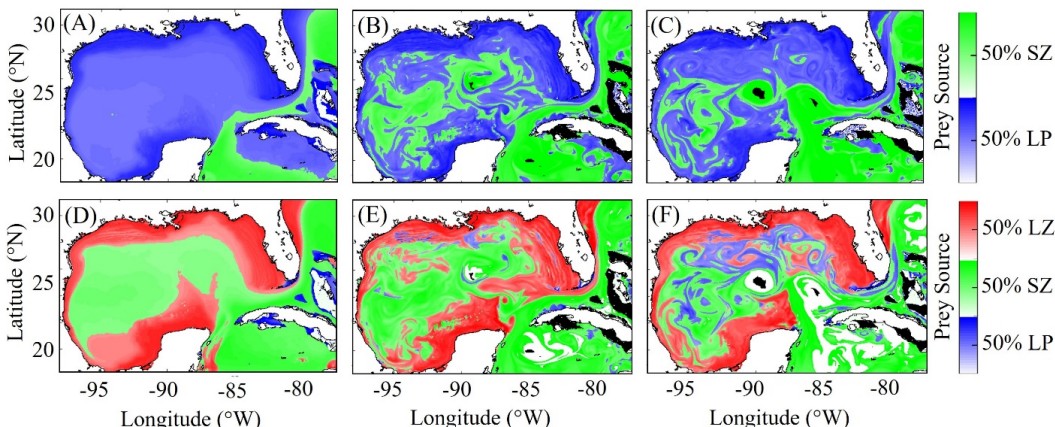


**Figure 8 (A-F):** Dominant prey source for simulated large zooplankton (LZ, top) and predatory
zooplankton (PZ, bottom). Colors indicate which prey are dominant. Brightness indicates percent
of the dominant prey in the zooplankton diet. Annual averaged field for LZ (A) and PZ (D).
Instantaneous winter condition for LZ (B) and PZ (E) on simulated day 4 February 2012.
Instantaneous summer conditions for LZ (C) and PZ (F) on 4 August 2011.

To investigate which prey source contribute the most to LZ and PZ diets, we computed each prey
source term for both LZ and PZ at each grid cell (**Fig. 8**). As we would expect, the dominant prey
source for LZ and PZ closely aligns with the spatial variability in their respective trophic positions.
For LZ diet, herbivory dominates throughout the GoM, except for the Loop Current (**Fig. 8A**). The
LP contribution to LZ diet is highest on the shelf, where LP biomass is also high due to the
competitive advantage LP have over SP in high nutrient conditions. In contrast, PZ diet varies with
the relative availability of SZ and LZ prey. In the oligotrophic region, PZ feed mainly on SZ
(heterotrophic protists), because LZ biomass is relatively low. On the shelf, they consume
primarily LZ (**Fig. 8D**). Despite the significant change in dominant prey between the shelf and
oligotrophic regions, PZ trophic positions remain fairly consistent (**Fig. 7D**) because SZ in the
oligotrophic region and LZ in the shelf region both feed predominantly on phytoplankton. In the
instantaneous fields for winter (**Fig. 8B, E**) and summer (**Fig. 8C, F**), the dominant prey for both
LZ and PZ show substantial mesoscale variability indicating that oceanographic features such as
fronts and eddies influence not only zooplankton biomass but also their ecological roles.

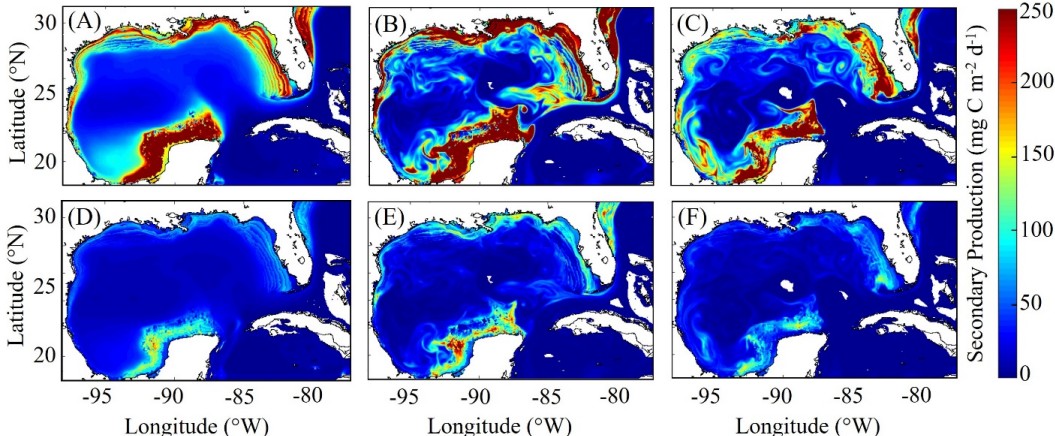

686

**Figure 9 (A-F):** Vertically integrated secondary production (mg C m$^{-2}$ d$^{-1}$) by simulated large zooplankton (LZ, top) and predatory zooplankton (PZ, bottom). Annual average of secondary production for LZ (A) and PZ (D). Instantaneous model output of secondary production in winter for LZ (B) and PZ (E) on simulated day 4 February 2012. Instantaneous model output for secondary production in summer for LZ (C) and PZ (F) on 2 August 2011.

To our knowledge prior to the current study the regional secondary production for the GoM has yet to be quantified. In terms of the entire GoM, we find that secondary production averaged $66 \pm 8$ mt C yr$^{-1}$ and ranged from a minimum of 51 mt C (in 1999) to a maximum of 82 mt C (in 2011). In the oligotrophic region, LZ secondary production averages $35 \pm 5$ mg C m$^{-2}$ d$^{-1}$ while PZ secondary production is $11 \pm 2$ mg C m$^{-2}$ d$^{-1}$ (**Fig. 9**). The annual secondary production minimum develops at the end of December while the annual maximum develops in the beginning of June (**Table 1**). In this region, mesozooplankton are responsible for $14 \pm 2$ mt C yr$^{-1}$, equivalent to 6% of NPP. In the shelf region, secondary production is about 4-fold higher, with LZ production of $146 \pm 17$ mg C m$^{-2}$ d$^{-1}$ and PZ production of $42 \pm 5$ mg C m$^{-2}$ d$^{-1}$. Here, the annual minimum also develops at the end of December while the seasonal maximum occurs near the end of July (**Table 1**). Secondary production in the shelf region averages $51 \pm 6$ mt C yr$^{-1}$ and is equivalent to 13% of NPP.

**4      Discussion**





Many parameters in biogeochemical models are poorly constrained by observations and laboratory
studies and/or highly variable in the environment. The numbers and uncertainties around these
parameters allow PBMs with varying degrees of tuning to reproduce a single ecosystem attribute
(e.g., surface Chl) even if multiple processes are inaccurately represented (Anderson, 2005;
Franks, 2009). Once validated, one of the main values of coupling physical and biogeochemical
models (i.e. PBMs) is their utility in making inferences about portions of the lower trophic level
that are under sampled and/or difficult to measure in the field. If PBMs are to be utilized for
explaining variability rather than just fitting an observational dataset, multiple ecosystem attributes
must be validated and the underlying model structure and assumptions critically evaluated. In the
section below, we further justify changes to model structure by evaluating the underlying
assumptions in default NEMRUO and discuss model-data mismatch before drawing conclusions
on the GoM zooplankton community and the implications of its dynamics on higher trophic levels.

### 4.1    Justification for NEMURO modifications

The phytoplankton community in the North Pacific (NP) domain where NEMURO was originally
designed is largely composed of nanoplankton (i.e. original SP) and microplankton (i.e. original
LP). By default, SP are assumed to represent coccolithophores and autotrophic nanoflagellates,
which can be important prey of copepods and other mesozooplankton in temperate and subpolar
regions (Kishi et al., 2007). However, in tropical regions such as the GoM, smaller
picophytoplankton taxa typically dominate particularly in highly oligotrophic regions. Common
picophytoplankton found in the GoM include cyanobacteria and picoeukaryotes which are too
small for most mesozooplankton to feed on. Consequently, the SP to LZ grazing pathway was
removed in the model. We found that removal of this grazing pathway allowed the model to
simulate a more realistic phytoplankton community in the shelf region. Despite intuition, SP
largely dominated the shelf region in the model when LZ were allowed to graze on SP. After closer
inspection we found that grazing of SP sustained LZ biomass on the shelf to levels where top-
down pressure constrained LP standing stocks. This prevented large blooms of LP leading to a
competitive advantage for SP even in highly eutrophic conditions (e.g. near the Mississippi river
delta). We found this was true under a wide range of LP maximum growth rates, LP half saturation
constants, and LZ/PZ grazing rates. Thus, removal of SP to LZ grazing pathway added ecological
realism and improved the model solution.





During the model tuning process, we also found that despite a wide range of tested parameter sets
the model (with default quadratic mortality formulation) was unable to simulate mesozooplankton
biomass low enough to match SEAMAP observations in the oligotrophic region. Even with
unrealistically low phytoplankton biomass, equivalent to approximately 50% of surface Chl
observed in SeaWIFS images, the model overestimated mesozooplankton biomass. We found that
to achieve realistic levels of mesozooplankton biomass in the oligotrophic region, default LZ and
PZ mortality parameter values needed to be increased by an order of magnitude. However, this
produced unrealistically high loss rates in the shelf region leading to mesozooplankton biomass
estimates that were substantially lower than SEAMAP shelf observations. Implementation of
linear mortality on all biological state variables (except PZ) resolved this issue by providing the
model with greater dynamic range. In NEMURO, and other biogeochemical models, quadratic
mortality is often used to increase model stability and/or is mechanistically justified as representing
the impact of unmodeled predators that co-vary in abundance with prey (Gentleman and
Neuheimer, 2008; Steele and Henderson, 1992). However, grazing losses of all state variables
(except PZ), are already explicitly modeled in NEMURO by default. Hence, removal of quadratic
mortality also added ecological realism and improved the model solution. Quadratic mortality was
retained for PZ, to account for the implicit predation pressure of un-modeled planktivorous fish.

### 753     **4.2    Model-data mismatch**

The PBM in this study captures a wide range of key regional ecosystem attributes across multiple
trophic levels. Surface Chl estimates were found to agree closely with satellite measurements,
reproducing patterns in both the oligotrophic and shelf region. The latter of which, apart from the
northern shelf, has not been well resolved by previous PBMs (e.g., Gomez et al., 2018; Xue et al.,
2013). The lack of a shelf Chl signature in previous studies may, in some cases, be overly attributed
to bias in satellite measurement due to high concentrations of colored dissolved organic matter on
the shelf. While a clear shelf signature is resolved in the NEMURO-GoM, we find greater model-
data mismatch on the shelf compared to oligotrophic regions. This is an expected finding when
considering the model incorporates climatological river forcing while actual variability is in reality
much more complex. Benthic processes that are not included in the NEMURO-GoM, such as
denitrification (Fennel et al., 2006), may also contribute to model-data discrepancies in the shelf
region.





The most noticeable surface Chl model-data mismatch occurs in the southern GoM on the Campeche Bank (CB) where the model consistently overestimates surface Chl. This overestimation was also notably present in the PBM implemented by Damien et al. (2018) for the GoM, particularly in winter. We believe this discrepancy is driven by a combination of error in the hydrodynamic model associated with overestimation of shelf mixing and simulated nitraclines that are too shallow, which allows for unrealistic mixing of nitrate into surface waters. Nitrate profiles from the oligotrophic GoM during May 2017 and 2018 cruises (A. Knapp, pers. comm.) revealed concentrations are typically below detection limits at depths shallower than 100 m. However, nitracline depths estimated by the model were shallower than observed with an upper limit of approximately 80 m (DCM depth was ~100 m) in summer months. While this discrepancy has minimal impact on average surface Chl over most of the domain, significant model-data mismatch arises in persistent upwelling areas such as north of the Yucatan Peninsula. In this region, strong upwelling produces a thin filament of high Chl water that extends northward as frequently observed in satellite images. To the west, circulation on the CB is characterized by a westward flow. Together with the shallower simulated nitracline depths, we believe the regional circulation supplies the CB with excessive nutrient-rich water leading to an overestimation of Chl by the PBM.

We found the model-data mismatch on the CB was reduced in parameter sets that produced nitracline depths down to 100 m. However, these parameter sets were less realistic in other ways (e.g. improbably deep DCMs). Given the strong thermal stratification and depth of the nitracline found in the GoM, we believe nitrogen fixing cyanobacteria may be another important source of new nitrogen (other than upwelling and mixing) supporting the surface phytoplankton community in the GoM. In the process of model tuning, we noticed that increasing the DON pool by increasing the PON to DON decomposition rate was necessary to maintain both relatively deep nitraclines and realistic surface Chl by providing a slow leeching of ammonium near the surface through bacterial communities. The need for this slow production of ammonium in surface layers may reflect the importance of nitrogen fixation, which is not included in NEMURO (Holl et al., 2007; Mulholland et al., 2006). In future studies including diazotrophs as a separate phytoplankton functional type would be valuable to investigate the importance of nitrogen fixation in the GoM.



Novel to this study, model estimates of mesozooplankton biomass were shown to agree closely
with observations on the shelf and in the oligotrophic GoM. To our knowledge, this study provides
the first quasi regional zooplankton biomass model-data comparisons in the GoM along with the
first model-data comparisons of size-specific zooplankton biomass and grazing rates. Such
comparisons provide the first insights into the potential biases of traditional biogeochemical
models pertaining to zooplankton dynamics (Everett et al., 2017). While the PBM shows broad
agreement with zooplankton observations, some model-data mismatch occurs, particularly for LZ
grazing rates. Some of this discrepancy may arise from temporal sampling issues (rate
measurements were only available for May 2017 and May 2018) or from inaccuracies in the field
grazing measurements.  Due to phytodetrital aggregates and *Trichodesmium* colonies in the
zooplankton net tows, our in situ gut pigment measurements were based solely on phaeopigment
content.  True grazing rates were likely underestimated because undegraded Chl can be abundant
in the foreguts of zooplankton. An additional source of model-data discrepancy arises from the
fact that the NEMURO model formulation of LZ and PZ does not necessarily reflect a size class
of mesozooplankton, but rather reflects a functional type of mesozooplankton. In reality, there is
overlap between taxonomic groups with different functional roles and different sizes.
Since most PBMs focus on validating against satellite-observed surface chlorophyll, the dynamics
of the DCM is often insufficiently investigated. Consequently, many models predict DCM depths
that are far too shallow.  Identifying this issue in the literature proved to be difficult seeing that
most studies don't provide profiles of simulated Chl. We note that DCM depths in the DIAZO
model (Stukel et al., 2014) were often quite shallow or completely nonexistent in the portion of
the domain that included the oligotrophic GoM region. Underestimates of DCM depth in the
unmodified COBALT biogeochemical model has also been identified (Moeller et al., 2019). In our
investigation of (Gomez et al., 2018) we found that DCMs in the oligotrophic region were
commonly shallow and weak. In the default NEMURO simulation, DCM depths in the
oligotrophic region were typically at a depth of 25 m, which is much shallower than SEAMAP
observations in the region (80 $\pm$ 25 m). While this issue may seem insignificant, particularly if a
study is focused on mixed-layer dynamics, accurate placement of the DCM can have profound
impacts on PBM behaviors, because the DCM is typically collocated with the nitracline.
Unrealistically shallow DCMs and nitraclines permit unrealistically high nitrate fluxes into the
surface layer following mixing events.  Indeed, we believe that a slight underestimation in



nitracline depth near the Yucatan Peninsula in our model contributed significantly to the model
overestimation of surface Chl on the Campeche Bank.
For these reasons, we devoted substantial effort to tuning phytoplankton dynamics at the DCM.
Modifications to α (the slope of the photosynthesis-irradiance curve) and attenuation coefficients
allowed us to move the DCM down to realistic depths. However, an additional issue was present
in the default NEMURO simulations, the NEMURO-GoM, and every simulation that we
attempted. In all simulations that formed DCMs, the location of the DCM was always collocated
with a maximum in phytoplankton specific growth rate. However, our field measurements of
phytoplankton growth rates and NPP were either relatively constant with depth or declined in the
DCM. This is not surprising, given the low photon flux at the base of the euphotic zone and the
energetic demands required to upregulate cellular density of light harvesting pigments. However,
in traditional PBMs high biomass DCM cannot form with a low growth rate, because specific
mortality rates tend to co-vary with biomass even if (as in our model) quadratic mortality is not
included.
Phytoplankton mortality (in the model and in the observations) is dominated by zooplankton
(particularly protists). Since zooplankton abundance covaries with phytoplankton abundance and
zooplankton specific grazing rates increase with increasing phytoplankton abundance, specific
mortality must co-vary with abundance. This means that phytoplankton mortality rates must be
higher at the DCM biomass peak than in the surface layer and thus a DCM can only be maintained
if growth rates are high. We tested multiple options to try to maintain a DCM with low growth
rates, including using light-dependent grazing formulations (Moeller et al., 2019), but found no
parameterizations that could match the observations. We believe this DCM issue was responsible,
in part, for the overestimates of LZ grazing rates (**Fig. 4D**). The collocation of the biomass and
growth rate maxima also lead to substantial overestimates of production (particularly by LP) at the
DCM, which was then grazed by LZ. Future modeling studies should focus more effort on
dynamics of the DCM.
**4.3     Mesozooplankton dynamics in the open-ocean oligotrophic Gulf of Mexico**
Despite its nutrient-poor conditions, the open-ocean GoM ecosystem is a key region for spawning
and larval development of many commercially important fishes, including Atlantic bluefin tuna,



yellowfin tuna, skipjack tuna, sailfish, and mahi mahi (Cornic and Rooker, 2018; Kitchens and
Rooker, 2014; Lindo-Atichati et al., 2012; Muhling et al., 2017; Rooker et al., 2012, 2013). Why
so many species choose such oligotrophic waters as habitat for their larval stages is unknown, but
may be due to reduced predation risk (Bakun, 2013; Bakun and Broad, 2003). Regardless, rapid
growth and survival through the larval period depends on mesozooplankton prey that are suitably
abundant and appropriately sized for these larval fishes. These prey taxa may be especially
sensitive to increased stratification and oligotrophication associated with climate change, making
investigation of their dynamics and production an important topic of research.
Mesozooplankton biomass in the oligotrophic GoM was found to be strikingly low in both
observations and PBM estimates with approximately an order of magnitude less biomass in
comparison to the shelf. PBM results clearly show that this low biomass condition arises from
bottom-up resource limitation. Our results suggest that low phytoplankton biomass in oligotrophic
regions, and particularly within Loop Current Eddies, may even lead to localized and episodic
regions where mean concentrations approach thresholds for triggering collapse of
mesozooplankton grazing. Prey limiting conditions for mesozooplankton and their predators
would be expected to occur more frequently in the GoM during warmer ocean conditions. Higher
sea surface temperatures and increased thermal stratification could suppress vertical mixing,
resulting in lower phytoplankton biomass. Indeed, while NEMURO-GoM exhibits severe nutrient
limitation in surface waters, the nitracline in the model is actually weaker and shallower than in
situ measurements during our cruises (A. Knapp, pers. comm.). This suggests potentially greater
nutrient scarcity in surface waters than the model predicts.
Despite extreme oligotrophy and dominance of picophytoplankton, our model shows that both PZ
and LZ populations can be sustained at modest abundances in the oligotrophic GoM. Indeed, the
substantial abundances of large (>1-mm) mesozooplankton equivalent to 60% of total
mesozooplankton, as determined by both observations and model results (**Fig. 4A, C**) is an
important result that helps explain the success of larval fish in the region. Our results show that
large mesozooplankton (PZ) occupy a trophic position of approximately 3.0 in the open ocean
GoM, which is marginally lower than on the shelf where they feed primarily on small
mesozooplankton (LZ). This change in trophic position is associated with a switch from carnivory
to feeding predominantly on heterotrophic protists in the oligotrophic region. This result highlights





the importance of intermediate protistan trophic levels in sustaining mesozooplankton
communities in oligotrophic regions. Indeed, both LZ and PZ are found to ingest proportionally
more SZ in the open ocean than on the shelf.  Notably, these protistan trophic steps cannot be
quantified by routine field techniques because they have no pigment signature to make them visible
in gut pigment measurements and may not enrich in bulk $^{15}$N leading to isotopic invisibility from
a trophic perspective (Gutiérrez-Rodríguez et al., 2014). Despite their importance, they are also
often missing from GoM ecosystem models (e.g., Fennel et al., 2011) and severely
underrepresented or even absent in complex mass-balance constrained models (Arreguin-Sanchez
et al., 2004; Geers et al., 2016). (Arreguin-Sanchez et al., 2004; Geers et al., 2016). New insights
may arise from focused investigation of phytoplankton➜protist➜crustacean linkages in
oligotrophic regions in both model and experimental studies.   This will likely require the use of
next-generation technologies such as compound specific isotopic analyses of specific amino acids
that have been shown to enrich in protists (Décima et al., 2017) or DNA metabarcoding to assess
zooplankton gut contents (Cleary et al., 2016).
Another robust result of our model is the dynamic mesoscale variability in zooplankton abundance,
diet, and trophic position.  These results highlight the impact of Loop Current Eddies and
mesoscale fronts and other features in modifying the biogeochemistry and food web of the GoM.
The existence of hot spots of productivity in the GoM has been seen in observational studies (Biggs
and Ressler, 2001), and the importance of GoM mesoscale features to fish larvae has been
hypothesized (Domingues et al., 2016; Lindo-Atichati et al., 2012; Rooker et al., 2012).   Our
results suggest that these mesoscale structures may not only modify zooplankton abundances, but
also their trophic roles in the ecosystem, with implications for the transfer efficiencies of carbon
and nitrogen in the pelagic food web.
**5.0    Conclusions**
In this study, we used an extensive suite of in situ measurements to validate zooplankton dynamics
simulated by a PBM of the GoM. The model was able to capture broad patterns in phytoplankton
and mesozooplankton abundances, depth of the deep chlorophyll max, and growth and grazing
patterns.  However, a distinct discrepancy was found between vertical profiles of measured and
modeled growth rates of phytoplankton.  Despite testing multiple parameterizations for
phytoplankton growth and zooplankton grazing, no model solution was found that could simulate



a DCM with high biomass, but low growth rates.  Future research is needed to diagnose these
dynamical issues for the DCM.  Once validated, the PBM was used to investigate important
characteristics of the GoM mesozooplankton community. Our results suggest that small
mesozooplankton are largely herbivorous and large mesozooplankton largely carnivorous on the
GoM shelf. However, distinct changes in diet were noted in the oligotrophic GoM, where both
groups rely more on protistan prey.  Changes in diet and secondary production highlighted in this
study have the potential to impact food availability to higher trophic levels, such as pelagic larval
fishes. In future work, we plan to couple our model to an individual-based model of larval fish to
evaluate the extent to which food resources limit larval fish feeding and growth along their
transport pathways in the GoM. Insights from this ecosystem-based approach may help to better
resolve stock-recruitment relationship that are needed for sustainable fisheries management and
improved stock-assessment models.



*Code and data availability.*

The model code and model validation data used in this study can be downloaded from GitHub at https://github.com/tashrops/NEMURO-GoM. An idealized one-dimensional version of NEMURO-GoM written in Matlab is also provided. The three-dimensional NEMURO-GoM model outputs used in the study are available on the FSU-COAPS server in a Network Common Data Form (NetCDF format).

*Author Contribution.*

TAS conducted all numerical simulations and model analysis. EPC, SLM, and AB provided expertise on the hydrodynamic modeling. MRS and VJC provided expertise on the biogeochemical model coding and tuning. RS, MRL, and GZ processed and provided data that was central to NEMURO-GoM's validation. TAS wrote the manuscript with contributions from all authors.

*Competing interest.*

The authors declare that they have no conflict of interest

*Acknowledgements.*

We thank the captains and crew of the NOAA ship Nancy Foster and many of our colleagues from NOAA SEFSC and the NASA-funded Zooplankton from Space project. We thank Oliver Jahn for providing valuable direction in configuring the offline MITgcm package. We also thank Mandy Karnauskas and Sang-Ki Lee for their thoughtful advice and guidance on the project. This paper is a result of research supported by a grant from The Gulf of Mexico Research Initiative under the CSOMIO project, the National Oceanic and Atmospheric Administration's RESTORE Science Program under federal funding opportunity NOAA-NOS-NCCOS-2017-2004875, by a NOAA Fisheries and the Environment grant, and by NASA IDS grant #80NSSC17K0560.





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
