# Peer review of "Quantifying spatiotemporal variability in zooplankton dynamics in the Gulf of Mexico with"

_Biogeosciences, 2019_

## Referee Comment (RC1) · Anonymous Referee #1 · 5 Jan 2020

GENERAL COMMENTS

This manuscript is a concerted effort to address the fact that zooplankton have historically been under-studied in ecosystem models, which is problematic because of their key role for trophic transfers and biogeochemical cycling. The stated objectives are to develop and validate an existing model (NEMURO) for the Gulf of Mexico (GoM), and then to quantify mesozooplankton diet and secondary production. They describe their rationale for modifiying NEMURO, and assess the effect of these changes. They present model-data comparisons for biomasses and rates of both phytoplankton and

zooplankton, with misfits mainly related to mesozooplankton grazing and vertical profiles of phytoplankton growth. Their main insights were that herbivory/carnivory differed between mesozooplankton size classes on the shelf, and protists were an important food source in oligotrophic regions. They also estimated that secondary production was an order of magnitude less than new primary production.

The paper, which clearly required a substantial amount of work, is thorough in its undertakings and is written clearly. It makes novel contributions with respect to ecological modeling and GoM functioning. It is useful for other regions, as the authors provided code for their 1D and 3D models, and their Supplemental serves as a great example of decision-making required to configure a model. I recommend its publication, although as described below, some aspects could be fleshed out and some text could be streamlined.

SPECIFIC COMMENTS

1. Model development:

They made 5 structural modifications and changed 25 (but I counted 30 in Table S3) parameter values from the standard NEMURO model. Their modification procedure demonstrates how we can use and gain ecological insight in constructing models (e.g. pg 32 necessary slow leaching of PON to DON),and constitutes an example of best practice. It is a real contribution to our field, that is especially useful for new modelers.

With respect to the modifications: (1) They could arguably have left the SP to LZ connection and just set the SP prey preferences to be very low, but removing it entirely is not objectionable. (2) Replacing quadratic mortality with linear mortality for all biological variables except PZ is a sensible simplification, but should be supported by references to model sensitivity studies about this issue (e.g. Anderson et al., 2015) (3) Now using a "monotonic" ammonium inhibition term, I believe, refers to the fact that some inhibition functions result in nitrogen uptake that decreases as ammonium levels increase as well as exceeding the so-called maximum uptake rate. This has been noted

in the literature and many rational alternatives presented (e.g. Frost & Franzen, 1992, Vallina and Le Que ÌĄre ÌĄ 2008). It would be beneficial to have a sentence explaining what they mean by "monotonic", so that readers appreciate the problem and know to follow them in making the change. (4) Use of the Platt light limitation is commonplace, but it would be useful to state whether the original NEMURO model did or did not include photoinhibition. If it did, it would be helpful to explain why the Platt formulation is considered to be an improvement. (5) Replacing constant C:Chl with variable C:Chl is potentially the most critical of all their changes, since this variable dictates the modeled Chl levels, which are used to compare the model and data. While they do provide the equations in the appendix, a brief overview of how those equations work would be beneficial. Overall, as these are all obvious modifications, the text devoted to this in Sections 2 and 4 could easily be cut down.

2. Model Validation

The model-data comparisons state the different values, but this does not give the reader any sense on whether the differences actually matter. Any text they could add about potential ecological significance of these differences would be helpful.

As noted above, model-data comparison of Chl will be impacted by modeled C:Chl ratios. It would be useful to have a quantitative sense of the model uncertainty related to this model component, At the very least, the issue should be discussed somewhere.

They allege that their biggest model misfit to data is vertical profiles of phytoplankton growth, and were only able to achieve realistic DCMs by tuning multiple parameters. It is not surprising that vertical profiles are challenging to model correctly, as there are many sources of error in the vertical dimension. For example, vertical velocities are always uncertain because they are calculated from the continuity equation, and thus absorb any error in horizontal velocities. This paper's use of a constant vertical mixing coefficient is not representative of higher mixing at surface as compared to interior. Vertical profiles of light depend on attenuation coefficients, which in this model appear

to use Beers's Law, and thus are oversimplifications (e.g. see Anderson et al., 1993, 2015). None of these sources of error are mentioned in the text, but It would be helpful to identify them and therefore guide future investigations.

Their simulation always resulted in modeled DCM being collocated with the maximum growth rate, which is not supported by observations. On pg 34 line 841, they argue that growth rates at a DCM must be high to balance the high mortality at a DCM, a conjecture they based on their (unsupported) assertion that zooplankton abundance covaries with phytoplankton abundance. I question this assertion. Certainly many studies have generated high zooplankton with low phytoplankton and vice versa. Zooplankton abundance depends in part on top-down control (i.e. predators and/or closure scheme), and phytoplankton specific mortality depends on zooplankton abundance as much as it does on grazing. Thus, you could have low phytoplankton growth rates at a DCM and maintain it, because grazing pressure (ingestion X abundance) is not high if zooplankton abundance is low.

They found that grazing by small mesozooplankton was overestimated, whereas grazing by large mesozooplankton was underestimated, both by a factor of about 2 (pg. 22 and pg 33). I am skeptical about their explanation that this is due to modeled grazing reflecting functional groups as opposed to size classes, especially given that the two groups' biomasses were reasonably simulated. It seems far more rational to assume the problem lies with the characterization of grazing, especially since grazing formulations are known to be highly uncertain, particularly when considering ingestion of multiple prey types. Although they did assess the effect of changes to the maximum grazing rate, they do not appear to have examined the effect of the two other grazing parameters ( Ivlev coefficient and threshold), which is surprising given that these parameters would be expected to vary among taxa, and their NEMURO modifications were motivated by the "significant taxonomic differences that found between mesozooplankton communities and their prey in the GoM and the North Pacific". They also don't appear to have examined the effect of altering the mathematical form of the functional

response. I am not advocating that they conduct a slew of further sensitivity studies at this point (indeed, that could warrant its own publication), but I do think they should devote some text to describing issues with their modeled grazing, and the challenges of getting it right and the consequences of getting it wrong. Again, this will serve to encourage future studies of ways to better represent this critical process, and to test the effect of different candidate formulations.

3. Parameter Sensitivity

Sections 3.4 and S5 are dedicated to what they call their "Parameter Sensitivity" study. This is not a sensitivity study in the traditional sense, which assess the robustness of a model output to variations in parameters (e.g. +/- 10%). Rather it is more an examination of how the model output changes if some subset of the original NEMURO formulations/values were used. It is not obvious what the reader was supposed to learn from this comparison beyond what was already presented with the parameter tuning. It didn't help that there was no text to clarify Table S3 or that the Taylor/Target diagrams that were hard to see and were also not described in the text. I think this analysis could easily be removed without any loss to the paper. If the authors feel differently, I strongly recommend that they revise these sections to make the value of the comparison clear. I also suggest changing the name of the section to something more representative of what is covered (e.g. "Ecological Effect of NEMURO modifications")

4. Simulated mesozooplankton diet

Their use of trophic level was a clever way to summarize zooplankton diet information. However, their findings that, on the shelf, LZ is more herbivorous and PZ is more carnivorous, and that protists are more important diet source in oligotrophic are not unexpected since they prescribed their model to have that output. For example, LZ grazing has no terms for interference by other prey, i.e. any consumption of SZ does not reduce ingestion of LP. In contrast, PZ has 3-fold higher grazing rate on LZ, as well as ingestion of zooplankton interfering with ingestion of phytoplankton. Given that

diet was a fundamental aspect of their study, I think it would behoove them to add text about what these grazing functional responses are assuming with respect to prey preferences (e.g. see Gentleman et al., 2003).

5. Simulated secondary production

Despite this being a stated objective, secondary production was only given a paragraph in the text, and is not mentioned in the discussion/conclusions. This is a missed opportunity to gain some more insight into the GoM function. If the authors don't want to flesh out the analysis, I suggest removing it from the objectives and possibly also from the results.

TECHNICAL CORRECTIONS

In Methods, on pg 9 of manuscript say 3 observational benchmarks, but in S3 pg 3 say and list 5.

In S3 pg 6, should give units for alpha.

I realize writing parameter tables and equations can be challenging to proofread, but in my surface examination I noted some typos and would recommend authors carefully review. e.g. –Supp pg 11, KSi says ammonium half saturation constant, should say Silica –pg 18 under Limitation terms, two eqns for GL_SP2SZ, one should be for GP_LP2SZ

Table S3 would be improved if the percentage difference was listed

Intro (pg 3) missing point that mesozooplankton also exert top down pressure on protists, which indirectly affects phytoplankton biomass

Methods (pg 6) Steele and Frost, 1977 not good reference here, as very detailed grazing formulation uncharacteristic of early NPZ. Better reference is Steele, 1974.

Methods (pg 13) Note that this size-based definition for small zooplankton includes early stages of mesozooplankton, which may sometimes be prey, but generally mature
to mesozooplankton (i.e. there is a mismatch between the data and model characterizations). You may want to note this in the text, as that could contribute to model-data differences.

Methods (pg 15) Line 398 accent on second Decima

Results (pg 18). Table 1 does not appear to be related to data/model comparison, or validation. Should this perhaps be removed or at least moved to a separate section? If it is retained, it would be worth pointing out the interesting result that the max in the oligotrophic regions is less than the min on shelf.

Supp: RLP in Table S2 not highlighted as changed but different value, same as tauSP2SZ

Discussion pg 36. State that protists are often missing from models, despite their importance. However, many modern ecosystem models have both microzooplankton and mesozooplankton components. It would be worthwhile to reference some of these other models, as well as the substantial body of work related to appropriate ways to model them, esp. for mixotrophs (see Flynn and Mitra).

The last two sentences of the conclusions are not really conclusions, they are more future work and could be moved to the discussion or removed.

---

## Referee Comment (RC2) · Anonymous Referee #2 · 9 Jan 2020

GENERAL COMMENTS

In their manuscript, Shropshire et al develop an offline coupled physical-biogeochemical model to quantify zooplankton dynamics in the Gulf of Mexico (GoM). This study has 2 part: (1) the development, tuning and extensive validation of the biogeochemical model (adapted from NEMURO) and (2) the analysis of zooplankton dynamics, trophic pathways and their controlling factors on the shelf and in offshore waters. The manuscript is well written and the conclusions supported by the results. The authors carried out a significant validation effort, in particular on the zooplankton

side of the model, which is usually unconstrained. The validation also includes phytoplankton growth rates and primary production. In that respect, the study clearly stands out. The validated model is then used to provide insights on mesozooplankton biomass and diet on the shelf and in the open Gulf. Despite the large effort, the model validation has some limitations, detailed below, which should be better acknowledged. The model results on mesozooplankton diet, which represent the main scientific result of the study, is only a small of the manuscript and could be expanded, in particular with respect to their ecological meaning (e.g. trophic efficiency, fisheries), to get a better balance between model development and scientific findings in the manuscript.

SPECIFIC COMMENTS

The authors adapted NEMURO to the specificities of the GoM using an empirical method with parameter tuning and a parameter sensitivity analysis. The empirical tuning is carried out with a 1D (offshore) and the 3D version of the model. The procedure is detailed extensively in the supplement. The tuning is fairly qualitative and does not use statistics to minimize the model-data mismatch or at least this was not mentioned. For instance it is indicated that DCM magnitude is used as a benchmark during the tuning but observed DCM biomass was not available. This suggests that there was no systematics comparisons with observations during the process. Clarifications on the procedure could be added. Also, more comparison with the literature and from model/data comparison could be provided to support the choice of parameters. Although Text S3 is too long, some of the tuning explanations could be incorporated to the manuscript as it seems relevant to the reader. For instance a schematic of NEMURO including the default and modified pathways and parameters would be useful for the reader to get a general sense of the modifications. Other technical information could also be added to the manuscript. For instance, a brief summary (in a sentence or two) of the model forcing, initial, and open boundary conditions in the manuscript would be useful (L188-189). It would also be informative to have a table in the manuscript showing the parameters used in the sensitivity analysis and their range (L415-416).

Comment on the lack of useable chlorophyll concentration data in the SEAMAP cruises (L351-352, L463-472). Vertical profiles of chlorophyll concentration would have been very useful for the validation. Chlorophyll samples were not collected during the SEAMAP cruises? The authors could look into the World Ocean Database (WOD) and the World Ocean Atlas (WOA), the latter being used for the 1D model. A comparison of monthly averaged vertical profiles of chlorophyll in WOD and in the model would be informative since the magnitude of the DCM cannot be validated with the SEAMAP data. Alternatively an average profile in the offshore area could be compared with a spatially averaged profile from WOA. This would provide a better comparison of the vertical distribution of chlorophyll.

As mentioned above, the validation is thorough but there are some mismatches that are not properly acknowledged. For instance there is a large discrepancy between observed and simulated surface chlorophyll on the shelf (Fig. 2F) but it is referred to as a good agreement in the manuscript (L419, L755-756). The model clearly overestimates surface chlorophyll on the shelf, which may have implications for the secondary production. This should be discussed. Also, regarding the model-data mismatch in grazing rate (L546-549), the statement that it is due to the model functional types not reflecting the size classes would be true only if the total grazing was similar in the model and observations, which is not the case for mesozooplankton. A better explanation should be provided. Specific growth rates at the DCM are discussed during the validation (L556-557) but the data are not presented. A figure of the specific growth rate profiles could be added to the supporting material? Regarding the "sensitivity" analysis, model results are compared with observations (e.g. Figure 6). Some model-model comparisons could also be presented. For instance, mesozooplankton biomass cannot be evaluated because the observations are depth averaged (L595-597) but this information is available in the model and therefore the effect of parameter/functional changes can be evaluated with the model.

The benthic condition in the model is simple and more information supporting this

choice could be added (L167-170). For instance it is not clear what is the basis for the 10% conversion of particulate matter. The fact that this choice "had no significant impact on the model" should be explained. The authors also note that benthic processes such as denitrification are not included in their model (L763-765), which may contribute to model-data discrepancies on the shelf. Including denitrification would probably result in lower phytoplankton biomass, hence a better agreement with observations. It might also alter the trophic pathways or at least the magnitude of the fluxes. Therefore, I wonder why this important process for shelf regions was not included, as it is relatively easy to implement (e.g. Fennel et al, 2006). The potential impact on the model results should be discussed.

Model validation and results are discussed with respect to the open Gulf (>1000m) and the shelf (<1000m). This choice seems reasonable. There are clearly strong differences between these 2 regions, which are highlighted in the validation and analysis of the results. However these two regions are not uniform and therefore averaging may be misleading. The open Gulf region is relatively uniform, although as noted in the manuscript biological dynamics are different in/out of the Loop Current and eddies. The GoM shelf area includes the western Florida shelf, the northern GoM shelf, the western GoM shelf and the northern Yucatan shelf, which have different characteristics. These differences are relevant for the study. Presenting model validation for the different sub-regions would be very informative. For instance the surface chlorophyll mismatch may occur on the Yucatan shelf only or due to a poor representation of the northern Gulf. Such subdivision was used recently by Gomez et al (https://doi.org/10.5194/bg-2019-430). Discussing the results regionally would also provide more in-depth information on zooplankton dynamics in the Gulf and give more confidence to some of the results. For instance, even though the Yucatan shelf results are questionable, they would not influence the other shelf areas.

TECHNICAL CORRECTIONS

L30: does mt stand for metric tonne? if yes please use t instead or alternatively 10ˆ6 g

throughout the manuscript

L187: "in Supplement S3"

L486-493: why switching to carbon units? mezozooplankton biomass is in mmol N m-3 in Figure 3.

L512: "sampled yearly": you mean a climatology? Could you rephrase?

L517: replace "community" by "biomass"

L558: replace "estimated" by "simulated"

L568-561: I don't understand these two statements. Why are you referring to all bio-geochemical models?

L715: "NEMURO"

L711-713: PBMs are not used to fit an observational dataset and not just meant to explain variability. You may want to rephrase your statement.

L744-745: can you provide some support from the literature for this choice?

L785-794: Is this related to the CB mismatch? If not start a new paragraph

L890: "Despite their importance in the offshore region"

L891: Above you mention the "importance of protists in oligotrophic regions" but then you cite Fennel et al (2011), which is a northern GoM shelf model.

L892-893: Remove the double citation

Figure 2: Why having twice the same information? panels A and B should be removed, they could also be replaced by a 1:1 chl plot and a bias map

Figure 3: Same comment as for Figure 2. Figure 3a: A line plot would be better to compare the model with observations Figure3b-e: What is the depth range used for averaging? it is depth averaged right, not integrated?

Figure 6: It is difficult to assess the individual sensitivity experiments because they are all represented as black dots. Can you color-code them or plot experiment numbers rather than dots?

Figure 8: It is a bit difficult to get around the color scale. The information is also is also somewhat redundant with Figure 7.

Supplement:

- for model parameters/processes you could use NO3 and NH4 instead of NO2 and NH2

- the KSI line in Table S2 indicates Ammonium half saturation constant (mmol N m-3)

---

## Referee Comment (RC3) · Anonymous Referee #3 · 9 Jan 2020

Shropshire et al. used the NEMURO biogeochemical model (Kishi et al., 2007) to describe spatiotemporal patterns in zooplankton biomass across the Gulf of Mexico (GoM). They made a series of changes to the original NEMURO formulation and parameterization to represent better low trophic level dynamics in the warm waters of the GoM. They used the MITgcm offline tracer advection package, which allows running the biogeochemical model offline, using existing model outputs from ocean circulation models. This significantly increases the model time step compared to online-coupled models, thus reducing the required time for model simulations.

[Figure]

Comparisons between observed and simulated patterns of zooplankton biomass show good agreement. Also, there is a good correspondence between simulated and observed surface chlorophyll patterns in the open GoM. However, the model does not reproduce realistic coastal chlorophyll patterns at surface, and important differences between model and observed rates for grazing, primary production, and specific growth are reported.

I have the following main concerns that need to be addressed before I can recommend publication.

1) A validation for nutrient's patterns should be included to gain additional confidence in the model results.

2) The model is not able to reproduce surface chlorophyll patterns in the coastal regions (bottom depth <1000 m). Is this a consequence of the model parameterization or a misrepresentation of river runoff fluxes? Assuming that all rivers have the same nutrient concentration than the Mississippi river is wrong. The USGS have data that should be considered to better constrain the land-ocean nutrient fluxes.

A mean time series for all the coastal regions in the GoM does not seem to me appropriate, because there are important differences among shelf regions. I would suggest an independent comparison for the shelves off Louisiana-Texas, Mississippi-Alabama, west Florida, and Yucatan.

3) The authors reported a good correlation between model and observed vertical profiles of chlorophyll. But a good correlation not necessarily implies that the model is simulating well the concentration values. A figure showing chlorophyll vertical profiles should be included in the paper main body or the Supplement, ideally displaying data for each season.

4) The in-situ data used for model validation was mostly based on measurement collected in two cruises during May of 2017 and 2018, with all the cruises stations located

in the open GoM. These data do not allow evaluating whether the model is reproducing cross-shore patterns or seasonal variability. There is abundant data in the northern GoM that the authors could use to improve the model validation, like the Gulf of Mexico coastal ocean observing system (GCOOS).

5) The original Kishi et al model included a small, large and predatory zooplankton to represent ciliates, copepods (mesozooplankton), and euphausiids (macrozooplankton). Shropshire et al. redefined large and predatory zooplankton as two size-classes of mesozooplankton. This conceptual change is not indicated in Section 2.1.2. Do the zooplankton parameters in the model, like maximum growth rate and maximum grazing rate need to be revisited after this redefinition? I dislike that the model validation results in Section 3 can be dependent of the size-class arbitrary choose to define large and predatory zooplankton.

———————————————————

---

## Author Comment (AC2) · 6 Feb 2020

The comment was uploaded in the form of a supplement:
https://www.biogeosciences-discuss.net/bg-2019-463/bg-2019-463-AC2-supplement.pdf

---

## Author Response (AR1)

*The author's thank the referees for their insightful and detailed comments. For convenience we have kept our* *original responses in blue* *and the* *revisions in red*.

**Response to Anonymous Referee #1**

**1. Model development**

Replacing quadratic mortality with linear mortality for all biological variables except PZ is a sensible simplification, but should be supported by references to model sensitivity studies about this issue (e.g. Anderson et al., 2015)

*We thank referee #1 for the suggestion and will include the citation in the revised manuscript.*

*We have now included the citation in the revised manuscript (line 208).*

It would be beneficial to have a sentence explaining what they mean by "monotonic", so that readers appreciate the problem and know to follow them in making the change.

*We reference the "non-monotonic" behavior of the default ammonium inhibition term to describe the result that at high nutrient concentrations total nutrient uptake by phytoplankton (i.e. uptake of $NO_3$ + uptake of $NH_4$) can actually decrease despite increases in the total nitrogen available to phytoplankton. This is unrealistic and hence was replaced with a more commonly used ammonium inhibition term. We will make this clearer in the revised manuscript.*

*We have now more clearly explain the "non-monotonic" behavior of the default ammonium inhibition term where we state: "At high $NO_3$ concentrations, the default term is known to generate unrealistic phytoplankton nutrient uptake patterns in which total nutrient uptake (i.e. uptake of $NO_3$ + uptake of $NH_4$) can actually decrease despite increases in $NH_4$ (and constant $NO_3$)" (lines 219-222).*

Use of the Platt light limitation is commonplace, but it would be useful to state whether the original NEMURO model did or did not include photoinhibition.

*The original NEMURO light limitation functional form does includes photoinhibtion implicitly. Platt functional form was used because it allows one to explicitly control the amount of photoinhibiton. Additionally, the Platt functional form is now more commonly used and thus parameter values were easier to find for comparison (e.g. initial slope of the PI curve ($\alpha$)). We will acknowledge this in revised manuscript.*

*We have now included a sentence letting the reader know that while NEMURO light limitation formulation includes photoinhibtion it is only implicitly represented (lines 223-224).*

Replacing constant C:Chl with variable C:Chl is potentially the most critical of all their changes, since this variable dictates the modeled Chl levels, which are used to compare the model and data.

While they do provide the equations in the appendix, a brief overview of how those equations work would be beneficial.

*In addition to having the description in the supplemental, we will include text in the revised manuscript dedicated to describing how the Li et al. (2008) C:Chl model equations work and the basis for them. For example, we will include a brief sentence on the original Geider et al. (1998) C:Chl model which is foundational to the Li et al. (2008) model.*

*Equations from the Geider et al. (1998) C:Chl model form the basis for the Li et al. (2008) model. We now include a brief description of the Geider model where we state "The Li et al. (2010) equations build on a previously constructed dynamic regulatory model of phytoplankton physiology which describes C:Chl variability under balanced growth and nutrient saturated conditions at constant temperature (see Geider et al., 1998))." (lines 233-236).*

**2. Model validation**

The model-data comparisons state the different values, but this does not give the reader any sense on whether the differences actually matter. Any text they could add about potential ecological significance of these differences would be helpful.

*We acknowledge that there could be more discussion on the potential ecological significances of model-data mismatch to give the reader a better sense of how the model-data mismatch may influence dynamics in the model. We thank review #1 for this suggestion and will address the ecological significance of the main model-data mismatch in our discussion (e.g. surface Chl on the shelf and grazing mismatch).*

*We now discuss the ecological impact regarding the primary model-data mismatch in NEMURO-GoM. This includes text devoted to discussing the ecological impact of surface Chl discrepancies (lines 804-810), discrepancies involving the DCM (lines 843-848), and mesozooplankton grazing discrepancies (lines 882-899). To increase organization in the final manuscript we separated the discussion of this model-data mismatch into three subsections and have included the paragraph devoted to discussing ecological impact at the bottom of each section.*

As noted above, model-data comparison of Chl will be impacted by modeled C:Chl ratios. It would be useful to have a quantitative sense of the model uncertainty related to this model component. At the very least, the issue should be discussed somewhere.

*We agree that quantifying uncertainty in C:Chl ratios is important moving forward as future models will likely continue to depend heavily on satellite Chl for the bulk of the model validation. Unfortunately, there is a paucity of in situ C:Chl data from the Gulf of Mexico that can be used for validation. We have ensured that our estimated C:Chl values are reasonable given patterns found in similar ecosystems, but in the revised manuscript we will also emphasize the need for more in situ measurements of this property in the GoM.*

*In section (4.2.1) "Surface chlorophyll discrepancies" we highlight the need for in situ measurements of C:Chl to better evaluate model-data mismatches where we state: "Future PBMs will likely continue to depend heavily on satellite Chl for the bulk of model validation and hence more in situ samples are needed which resolve changes in phytoplankton light harvesting pigments*

*along gradients from coastal to oligotrophic regions and from the surface to the DCM. Without these observations it is difficult to investigate mismatch between model and satellite ocean color products or in situ profiles of Chl" (lines 775-779).*

This paper's use of a constant vertical mixing coefficient is not representative of higher mixing at surface as compared to interior. Vertical profiles of light depend on attenuation coefficients, which in this model appear to use Beers's Law, and thus are oversimplifications (e.g. see Anderson et al., 1993, 2015). None of these sources of error are mentioned in the text, but it would be helpful to identify them and therefore guide future investigations.

*The model does not use a constant vertical diffusivity but one that varies in space and time and is calculated in the mixed layer package of MITgcm using the KPP mixed layer parameterization which is commonly used in hydrodynamic ocean models. Vertical diffusivity was constant in the one-dimensional model during our tuning process. We acknowledge that Beer's law is an over simplification for the attenuation of light particularly in highly riverine influenced areas which may have high turbidity associated with suspended material and not with high Chl. We will note this in the revised manuscript.*

*In section (4.2.1) "Surface chlorophyll discrepancies" we have also acknowledged the importance of including a more realistic light attenuation formulation in the model where we state: "Implementing more realistic light attenuation (e.g. wavelength-specific light attenuation or inclusion of CDOM) could further improve estimates of phytoplankton biomass on the shelf as primary production can be sensitive to different light attenuation formulations (Anderson et al., 2015)." (lines 768-771).*

Their simulation always resulted in modeled DCM being collocated with the maximum growth rate, which is not supported by observations. On pg 34 line 841, they argue that growth rates at a DCM must be high to balance the high mortality at a DCM, a conjecture they based on their (unsupported) assertion that zooplankton abundance covaries with phytoplankton abundance. I question this assertion.

[Figure]

*Figure: Average model biomass profiles from a spatial average over a 2° x 2° sample box located in the oligotrophic GoM with an origin at 24° N, 91° W that shows high correlation of SZ and phytoplankton biomass.*

*We found that in all of our NEMURO parameter sets small zooplankton (SZ) biomass closely varied with phytoplankton biomass, and this follows fairly clearly from the differential equations used for protistan growth. At the low biomass conditions existing in the open GoM, protistan growth rates are almost a linear function of prey (small phytoplankton) biomass, while mortality scales with the biomass of large and predatory zooplankton. Increased phytoplankton biomass thus leads to increased small zooplankton biomass, until the increase SZ growth is balanced by increased SZ mortality. This leads to a strong correlation between SP and SZ with depth in the model. Whether there is a similar correlation in the field is unknown, because of the paucity of field studies that differentiate phagotrophic and non-phagotrophic protists. The attached figure shows typical profiles of SP, LP, SZ, LP, and LZ found in the center of the oligotrophic GoM. As a response to referee #2 we included a figure showing the higher specific rates at the DCM in the model while measured specific phytoplankton growth rates are lowest in the DCM.*

*We have removed the statement that model growth rates must be high at the DCM to balance high grazing by microzooplankon. Although we think this is common in biogeochemical models (based on our own non-exhaustive testing of other models) we now feel that it is peripheral for the paper. However, we have kept the discussion of shallow DCMs (lines 812-827) in PBMs and the suggestion of further investigation of DCM dynamics for future studies (lines 848-849).*

They found that grazing by small mesozooplankton was overestimated, whereas grazing by large mesozooplankton was underestimated, both by a factor of about 2 (pg. 22 and pg 33). I am skeptical about their explanation that this is due to modeled grazing reflecting functional groups as opposed to size classes, especially given that the two groups' biomasses were reasonably simulated. It seems far more rational to assume the problem lies with the characterization of grazing, especially since grazing formulations are known to be highly uncertain, particularly when considering ingestion of multiple prey types.

*We agree that errors in the grazing functional form could also be a source of error leading to the model-data mismatch. We will include this point in the revised manuscript and discuss the need for future grazing measurements in the field to help select among a group of grazing functions. However, we do believe that most of the issue with the balance in grazing between PZ and LZ is due to the fact that PZ in NEMURO is explicitly defined and parameterized as a group of higher trophic level zooplankton that can feed on LZ. In reality, while there is a correlation between size and trophic level in the ocean, there are many predatory zooplankton that are <1-mm and many suspension-feeding zooplankton that are >1-mm.*

*We have now included text addressing how errors in grazing formulation may be a source of model-data mismatch. We also highlight the need for future grazing measurements in the field to help select among a group of grazing functions where we state: "Uncertainties in model grazing formulations could also contribute to model-data mismatch (Gentleman et al., 2003a; Sailley et al., 2015). Future in situ grazing measurements are needed to enable an objective selection of grazing formulations and parameter values. In particular, field studies that shed light on prey*

*selectivity would be useful for parameterizing PBMs with multiple mesozooplankton functional groups, such as NEMURO-GoM."(lines 868-873).*

**3.  Parameter sensitivity**

Sections 3.4 and S5 are dedicated to what they call their "Parameter Sensitivity" study.
This is not a sensitivity study in the traditional sense, which assess the robustness of a model output to variations in parameters (e.g. +/- 10%). Rather it is more an examination of how the model output changes if some subset of the original NEMURO formulations/values were used… I think this analysis could easily be removed without any loss to the paper.

*We used this analysis to demonstrate the relative importance of changing parameters from default NEMURO, but now agree with referee #1 that this section is not necessary in the main text of the manuscript. In the revised manuscript we moved the analysis to the supplemental where we already have a table outlining the 18 parameter sets used in the analysis. Removal of this section also allows us to elaborate on some of the points put forth while not increasing the length of the manuscript further.*

*In the revised manuscript we have moved the parameter sensitivity section to the supplemental to section S5 "Parameter Sensitivity Experiments."*

**4.  Simulated mesozooplankton diet**

Their use of trophic level was a clever way to summarize zooplankton diet information. However, their findings that, on the shelf, LZ is more herbivorous and PZ is more carnivorous, and that protists are more important diet source in oligotrophic are not unexpected since they prescribed their model to have that output

*We agree that these results follow from the ways in which the zooplankton are parameterized, however these results are not immediately obvious a priori. While we certainly would have predicted a priori that LZ would feed predominantly on protists in oligotrophic regions and on large phytoplankton in coastal regions (and hence have higher trophic levels in the open GoM), it was not readily apparent that PZ would have lower trophic levels in the open ocean. Indeed, one might have expected a priori that PZ trophic level would follow that of LZ.*

*We have now addressed the need for future in situ grazing measurements to better parameterized PBMs that include multiple mesozooplankton functional groups (870-873).*

**5.  Simulated secondary production**

Despite this being a stated objective, secondary production was only given a paragraph in the text, and is not mentioned in the discussion/conclusions. This is a missed opportunity to gain some more insight into the GoM function. If the authors don't want to flesh out the analysis, I suggest removing it from the objectives and possibly also from the results.

*We understand the reviewer's interest in secondary production and plan to elaborate on this discussion in the revised manuscript. We also note that we are preparing a second manuscript*

*that adds an individual-based model of larval fish to explicitly investigate the relationship between secondary production and food availability and ultimately starvation for Atlantic Bluefin tuna larvae in the Gulf of Mexico.*

*We have expanded on secondary production in the revised manuscript. Specifically we now include results on secondary production in an outside of Loop Current Eddies (lines 674-692) as well as results on the relationship between secondary production and phytoplankton biomass (lines 663-673).*

**6. Technical Corrections**

*We appreciate the detailed corrections provided and have implement them accordingly.*

*We have included the technical corrections in the revised manuscript.*

**1. Specific Comments**

The tuning is fairly qualitative and does not use statistics to minimize the model-data mismatch or at least this was not mentioned. For instance it is indicated that DCM magnitude is used as a benchmark during the tuning but observed DCM biomass was not available. This suggests that there was no systematics comparisons with observations during the process. Clarifications on the procedure could be added.

*While some studies have used quantitative cost functions as an approach for objective model tuning, such approaches are often limited to a very restricted number of benchmark measurements (e.g., satellite-derived chlorophyll and NPP). It is not immediately clear how to implement such approaches when a wide array of expected ecosystem properties are being considered (e.g., we considered sea surface chlorophyll, NPP, relative ratios of large:small phytoplankton, DCM depth, DCM biomass, ratio of phyto growth to zooplankton grazing, mesozooplankton biomass, mesozooplankton size structure, protistan grazing, and mesozooplankton grazing). These measurements come with different distinctly different levels of confidence, number of measurements available, and in some cases (e.g., relative contribution of large phytoplankton) had to be based on intuition from other similar regions. For this reason, we relied on a semi-quantitative approach – we compared results of multiple simulations to all of our expected benchmarks to search for simulations that gave a reasonable fit in all ways, rather than searching for one that minimized a specific cost function. We will add text to the supplemental and briefly in the revised manuscript which is similar to our response here to further clarify our tuning process.*

[Figure]

*Figure (A & B): Observed Chl profiles from calibrated fluorimeter obtained during May 2017 cruise (A). Frequency of observed DCM depth (B).*

*With respect to the DCM depth and magnitude that the referee enquired about, 88 calibrated fluorescence profiles were obtained from our Gulf of Mexico cruise in 2017 and these were used for reference in tuning (i.e these profiles provided a target value with chlorophyll concentrations at the DCM of ~0.5 mg m$^{-3}$). We will clarify this in the supplemental. No measurements of LP abundance were available but was used as a baseline from what the authors would expect to find. All other target metrics used in the one-dimensional model tuning are averages from synoptic data sources (SeaWIFS – Surface Chl, SEAMAP – MZB, SEAMAP – DCM depth). We will add text in the revised draft to clarify this.*

*Motivated by the reviewer's comment we have included a separate section in the Methods titled "Biogeochemical model tuning procedure" that better explains the tuning process (lines 239-259). However, we have chosen to keep most of the tuning details in the supplemental.*

Although Text S3 is too long, some of the tuning explanations could be incorporated to the manuscript as it seems relevant to the reader. For instance a schematic of NEMURO including the default and modified pathways and parameters would be useful for the reader to get a general sense of the modifications. Other technical information could also be added to the manuscript.

*We will include more detail of the model tuning procedure and also include some text giving the reader an understanding of the sources from which we obtained our initial conditions and forcing.*

*We now feel that the addition of the section titled "Biogeochemical model tuning procedure" along with information in the supplemental provides the reader with sufficient information on our model tuning process. We have included a brief description of initial/open boundary conditions to provide more technical information (lines 186-194).*

Vertical profiles of chlorophyll concentration would have been very useful for the validation… A comparison of monthly averaged vertical profiles of chlorophyll in WOD and in the model would be informative since the magnitude of the DCM cannot be validated with the SEAMAP data.

*We agree that vertical profiles of Chl are useful to increase confidence in the model solution. When writing the manuscript we did not have access to CTD data from our 2018 cruise and so we relied on the extensive SEAMAP fluorescence profiles for validation of the DCM. Since then we have processed the 2018 CTD data. In our revised manuscript we plan on including an additional validation figure devoted to show point-to-point model-data comparisons of Chl profiles based on the samples collected during both May 207 and 2018 lagrangian process study cruises. This figure will also include a histogram of DCM depths from SEAMAP fluorescence profiles and simulated DCM depth at corresponding times and locations.*

*In our revised manuscript we have now added a section devoted to model-data comparisons involving vertical profiles of Chl and nitrate. For data details see section 2.3.3 "Observed vertical profiles of chlorophyll and nitrate" lines (372-390). For model-data comparisons see section 3.3 "Chlorophyll and nitrate profile model-data comparisons" (lines 503-524) and Figure 4(A-C).*

As mentioned above, the validation is thorough but there are some mismatches that are not properly acknowledged. For instance there is a large discrepancy between observed and simulated surface

chlorophyll on the shelf (Fig. 2F) but it is referred to as a good agreement in the manuscript (L419, L755-756). The model clearly overestimates surface chlorophyll on the shelf, which may have implications for the secondary production. This should be discussed.

*We agree that the model overestimates Chl on some portions of the shelf. However, the model does a significantly better job overall at resolving the coastal signature in comparisons to past studies. We acknowledge this in the manuscript where we state "While a clear shelf signature is resolved in the NEMURO-GoM, we find greater model-data mismatch on the shelf compared to oligotrophic regions. This is an expected finding when considering the model incorporates climatological river forcing while actual variability is in reality much more complex" – line 790. We acknowledge some text could be added discussing how this model-data mismatch will influence secondary production. Text will be devoted to this topic in the revised manuscript.*

*In response to comments by reviewer #1 along with this comment we have added discussion on the specific ecological impact of the model-data mismatch on the shelf. This includes text devoted to discussing the ecological impact of surface Chl discrepancies (lines 804-810), discrepancies involving the DCM (lines 843-848), and mesozooplankton grazing discrepancies (lines 882-899). We also provide some ideas for how future studies could better resolve the Chl variability on the shelf (lines 765-771).*

Also, regarding the model-data mismatch in grazing rate (L546-549), the statement that it is due to the model functional types not reflecting the size classes would be true only if the total grazing was similar in the model and observations, which is not the case for mesozooplankton. A better explanation should be provided.

*We believe that there are some subtle points here that need to be addressed, and we will include them in the revised manuscript. In detail: while there is model-data mismatch for total mesozooplankton grazing it is not entirely clear that this model-data mismatch is outside the inherent uncertainty associated with the gut pigment approach to grazing measurements. Specifically, after zooplankton consume chlorophyll the acid in their guts will begin to convert chlorophyll to phaeopigments, although this conversion is neither instantaneous nor complete. Ideally, gut pigment measurements should consider both chlorophyll and phaeopigments, however, the presence of chlorophyll in detrital aggregates that are collected during zooplankton net tows can lead to substantial contamination. For that reason, we have only included phaeopigment when computing mesozooplankton grazing, which leads to a likely underestimate. Furthermore, the approach assumes that any group of mesozooplankton has a constant gut throughput time, which of course is an oversimplification of the biology of complex organisms. Any mesozooplankton grazing measurements thus need to be treated with considerable uncertainty, hence it is not clear that the model is necessarily wrong in terms of its estimate of overall grazing (note that we are certainly not arguing that the model is right and the measurements are wrong, simply that within the uncertainty inherent to the field measurement approach it is not possible to conclusively determine that the model is wrong about total mesozooplankton grazing rates). However, the model is fairly clearly wrong about the proportion of grazing mediated by PZ and LZ. We believe that the error in this ratio is due to the fact that PZ in NEMURO is explicitly defined and parameterized as a group of higher trophic level zooplankton that can feed on LZ. In reality, while there is a correlation between size and trophic level in the*

*ocean, there are many predatory zooplankton that are <1-mm and many suspension-feeding zooplankton that are >1-mm. We will attempt to make this clearer in the revised manuscript. Motivated by this comment and comment from referee #1 we will include some text discussing the potential source of error from the grazing functions.*

*We have now included three explanations for why model-data mismatch exists in regards to mesozooplankton grazing: (1) temporal sampling discrepancy, (2) errors in field measurements, and (3) errors due to grazing formulation (lines 859-870). We have kept text discussing the potential errors due to issues with size and functional form but have focused this source of error to model-data mismatch regarding the ratio between grazing by PZ and LZ (lines 874-881).*

Specific growth rates at the DCM are discussed during the validation (L556-557) but the data are not presented. A figure of the specific growth rate profiles could be added to the supporting material?

[Figure]

*Figure (A-C): Simulated average chlorophyll profile in the center of the oligotrophic GoM (A). Comparison of simulated specific growth and grazing rates (B) to observed phytoplankton specific growth rates from dilution experiments conducted during May 2017 and 2018 lagrangian process study cruises (C). Final sample depth was always located at the observed deep chlorophyll maximum.*

*We will add a figure of specific growth rates from our 2017 and 2018 cruise in supporting material.*

*We now think this is unnecessary as we have decreased the discussion of this topic (at the suggestion of reviewer #1).*

Regarding the "sensitivity" analysis, model results are compared with observations (e.g. Figure 6). Some model-model comparisons could also be presented. For instance, mesozooplankton biomass cannot be evaluated because the observations are depth averaged (L595-597) but this information is available in the model and therefore the effect of parameter/functional changes can be evaluated with the model.

*A full parameter sensitivity analysis with parameter changes and functional form changes could be focused on in a future manuscript. Indeed, our offline modeling approach makes this type of detailed analysis possible. Table S3 outlines the results of 18 different parameter sets which can give the reader a sense of how our parameter value modifications influence simulated mesozooplankton biomass. Given the (already long) length of this manuscript, and referee #1's comment that the analysis could be removed entirely, we believe that it is not best to add additional analyses.*

*As mentioned in our response. A full parameter sensitivity study could be a standalone study. We have chosen to move our parameter sensitivity analysis to the supplemental in response to comments by reviewer #1.*

Including denitrification would probably result in lower phytoplankton biomass, hence a better agreement with observations. It might also alter the trophic pathways or at least the magnitude of the fluxes. Therefore, I wonder why this important process for shelf regions was not included, as it is relatively easy to implement (e.g. Fennel et al, 2006). The potential impact on the model results should be discussed.

*We agree that denitrification can be important on the shelf and that it could be implemented in future versions of the model. However, it is not part of the standard NEMURO model and was not necessary for us to implement it, because our focus in the study was on the open ocean regions of the GoM where zooplankton biomass is likely to be limiting to higher trophic levels. In the revised manuscript, however, we will highlight the fact that NEMURO does not include denitrification which can be important on the shelf.*

*We have now more clearly stated that our model does not include benthic processes (lines 163-172) and that this could be a source of model-data mismatch (lines 766-768).We have supported this decision in the revised manuscript where we state: "The inclusion of a more complex sediment diagenesis model (including denitrification) would have added further realism. However, our main focus was to evaluate zooplankton dynamics in the oligotrophic region where higher trophic levels that depend on mesozooplankton secondary production may experience food limitation and where benthic processes are negligible" (lines 168-172).*

The GoM shelf area includes the western Florida shelf, the northern GoM shelf, the western GoM shelf and the northern Yucatan shelf, which have different characteristics. These differences are relevant for the study.

*We agree that the Gulf of Mexico shelf has many distinct regions (and any one of these distinct regions could be the focus of another manuscript), however in this (already long) manuscript, we prefer to keep our focus on the open ocean regions. In our own analyses we found that although the model-data mismatch was significantly worse on the Campeche Bank relative to other shelf regions the ability of the model to capture the temporal variability was similar among shelf regions.*

*We have added text devoted to discussing potential "next steps" future studies could take to better resolve variability on the shelf. This includes: (1) daily (not climatological) river nutrient data*

*being prescribed in the model, (2) representing benthic processes, (3) including a more realistic light attenuation formulation, (4) use of more in situ C:Chl data to better evaluate the simulated phytoplankton biomass on the shelf (lines 765-779).*

**2. Technical Corrections**

*We appreciate the detailed corrections provided and have implement them accordingly.*

*We have included the technical corrections in the revised manuscript.*

**Response to Anonymous Referee #3**

**1. General Comments**

A validation for nutrient's patterns should be included to gain additional confidence in the model results.

*Unfortunately there are surprisingly few nutrient measurements from the GoM available in public databases. When tuning the model, two things that we paid attention to were the expected collocation of the nutricline and deep chlorophyll maxima, and the expected low nutrient concentrations in the surface layers of the open ocean. The model accurately represents these dynamics. However, given the already long length of this manuscript and our specific focus on mesozooplankton dynamics, we do not believe a figure focused on nutrient dynamics is warranted.*

*Motivated by the reviewer's comment we revisited the availability of nitrate measurements in the GoM. We found that WOD had a few nitrate profiles that overlapped with our simulation period. We have now included validation of nitrate using these profiles (Figure 4C). Data description can be found on lines 378-390 and model-data comparisons on lines 518-524.*

The model is not able to reproduce surface chlorophyll patterns in the coastal regions (bottom depth <1000 m). Is this a consequence of the model parameterization or a misrepresentation of river runoff fluxes?

*We note that although the model does not do a great job of reconstructing average surface chlorophyll temporal variability in the coastal regions, it actually does a very good job (compared to previously published models and thus a clear step forward) of estimating spatial patterns of chlorophyll, particularly on the shelf. As we have noted in the manuscript, the region where the model performs worst is the Campeche Bank near the Yucatan Peninsula, where the model consistently overestimates phytoplankton biomass. The model's issues with estimating temporal variability on the shelf stem primarily from three issues: 1) we use climatological river forcing, which has a disproportionate impact on nutrient supply to the shelf. 2) because the primary focus of the model was on the deep ocean GoM, when tuning we focused more on this region. For instance, we considered the non-diatoms in the model to represent cyanobacteria, rather than the nanoflagellates and dinoflagellates that would play important roles near the coast. 3) the noted (and substantial) errors in the Campeche Bank are disproportionately important to the temporal signal of <1000 m chl when averaged in the model.*

*We have added text devoted to discussing potential "next steps" future studies could take to better resolve variability on the shelf. This includes: (1) daily (not climatological) river nutrient data being prescribed in the model, (2) representing benthic processes, (3) including a more realistic light attenuation formulation, (4) use of more in situ C:Chl data to better evaluate the simulated phytoplankton biomass on the shelf (lines 765-779).*

A mean time series for all the coastal regions in the GoM does not seem to me appropriate, because there are important differences among shelf regions. I would suggest an independent comparison for the shelves off Louisiana-Texas, Mississippi-Alabama, west Florida, and Yucatan.

*We fully acknowledge that there are issues (as we noted above) with including only a single plot for all coastal regions. However, our focus is not on the coastal regions, and this is already a very long manuscript, so we do not believe that separate panels for each of the shelf regions is warranted.*

A figure showing chlorophyll vertical profiles should be included in the paper main body or the Supplement, ideally displaying data for each season.

*We agree that vertical profiles of Chl are useful to increase confidence in the model solution. When writing the manuscript we did not have access to CTD data from our 2018 cruise. Since then we have obtained this data. In our revised manuscript we plan on including an additional validation figure devoted to show point-to-point model-data comparisons of Chl profiles based on the samples collected during both May 207 and 2018 lagrangian process study cruises. This figure will also include a histogram of DCM depths from SEAMAP fluorescence profiles and simulated DCM depth at corresponding times and locations.*

*We have now included model-data comparisons of Chl (lines 379-383) in addition to model-data comparisons of DCM depth in the previous draft (Figure 4A & B).*

The in-situ data used for model validation was mostly based on measurement collected in two cruises during May of 2017 and 2018, with all the cruises stations located in the open GoM. These data do not allow evaluating whether the model is reproducing cross-shore patterns or seasonal variability. There is abundant data in the northern GoM that the authors could use to improve the model validation, like the Gulf of Mexico coastal ocean observing system (GCOOS).

*We appreciate the suggestion and investigated the GCOOS website but found no mesozooplankton grazing rate measurements available. In the manuscript we use the incredibly extensive SEAMAP data set for validating cross-shelf patterns, and believe that this is far more important for constraining mesozooplankton dynamics. However, as noted above, we have now included additional model-data comparisons for vertical profiles of Chl and nutrients.*

The original Kishi et al model included a small, large and predatory zooplankton to represent ciliates, copepods (mesozooplankton), and euphausiids (macrozooplankton). Shropshire et al. redefined large and predatory zooplankton as two size-classes of mesozooplankton. This conceptual change is not indicated in Section 2.1.2. Do the zooplankton parameters in the model, like maximum growth rate and maximum grazing rate need to be revisited after this redefinition? I dislike that the model validation results in Section 3 can be dependent of the size-class arbitrary choose to define large and predatory zooplankton.

*Originally PZ are defined in NEMURO as a group that includes gelatinous plankton and euphausiids. From the original Kishi manuscript "the biomass of the highest predator ZP is in a sense unrealistic in that it represents the total biomass of a number of species. We included ZP in NEMURO to get a more accurate representation of the biomass of ZL, which plays an important role in lower trophic ecosystems in the Northern Pacific, as well as to represent a suitable prey functional group for the higher trophic level linkages." We have treated LZ and PZ in much the same way, but in order to compare them to field data (and in a coming manuscript to the prey field*

*for larval Bluefin tuna) we have chosen to consider Kishi's LZ (mainly copepod) trophic group to be representative of mostly 0.2 – 1.0 mm organisms and Kishi's PZ (gelatinous predators and krill) to be representative of mostly >1.0 mm organisms.*

*We have now more clearly stated our size definition for mesozooplankton (lines 358-364) and also have more clear acknowledge the limitations of comparing size classes to functional groups (lines 874-881).*

---

## Author Response (AR2)

*The author's thank the referees for their insightful and detailed comments. We have provided our responses in blue.*

On line 778, they say "Without these observations, it is difficult to gauge mismatches between model and [data]". It is true that more in situ measurements would enable refinement of this component of the model, but there are two additional aspects that warrant mention. (a) The range of realistic values of C:Chl is high (e.g. 25-250). Therefore, a poorly modeled C:Chl ratio could result in a large misfit even when the modeled nitrogen/carbon-based ecosystem dynamics was excellent. (b) Quantifying uncertainty can be addressed in the absence of observations simply by conducting sensitivity tests on the components of the C:Chl submodel.

*We thank the reviewer for the additional suggestions. We have included these points in our discussion of C:Chl ratios where we state "In addition to validation, these measurements are needed to avoid erroneous model tuning. For instance, a model that exhibits significant mismatch with respect to surface Chl may in fact accurately estimate carbon based phytoplankton biomass while using unrealistic C:Chl ratios. One could arrive at incorrect conclusions about regional ecosystem dynamics as a result of modifying model parameters or structure in an effort to better fit Chl observations. Given the importance of C:Chl ratios in PBMs, future studies should quantify uncertainty in modeled Chl through sensitivity experiments focused on C:Chl model parameters and formulation, with explicit comparison to direct field measurements of phytoplankton C:Chl." (lines 790-797).*

The authors should be good role models and examine whether their modeled phytoplankton biomass (i.e. in carbon) is high.

*We agree that examining phytoplankton biomass in carbon (or phytoplankton C:Chl ratios) is important to conclusively determine if a model is accurately representing phytoplankton biomass. Unfortunately, there is a distinct paucity of true phytoplankton biomass (C or N) measurements in the Gulf of Mexico. In particular, we are not aware of any carbon based measurements in places like the Campeche Bank where the model shows its greatest Chl-based model-data mismatch. We are working with collaborators to try to get microscopy and flow-cytometry derived field measurements of phytoplankton carbon in the Gulf, but these are not yet available.*

I suggest they reference some of the problems with accuracy in modelled vertical diffusivities and currents that I mentioned in my original review.

*We agree that errors due to both biology and physics could be contributing to model-data mismatch at the DCM. We have now included this point where we state "Future PBM studies need to focus more effort on resolving ecological dynamics responsible for the formation of the DCM. Errors originating from the hydrodynamic model or use of temporally averaged velocity fields used in offline models may also contribute to model-data mismatch at the DCM. Vertical mixing is particularly important to PBMs, but is often poorly validated in hydrodynamic models. Greater coordination between physical modelers and biologists to constrain vertical fluxes should be considered an important avenue for improving PBM simulations moving forward." (lines 868-874).*

They could add a couple of sentences reminding the reader that modeling diet composition and secondary production relies entirely on the modeled prey selectivity formulation. They may also want to expand on this point to say that there exist a wide array of published multiple prey responses that have the potential to predict very different consumption rates. They should advocate for modeling studies that assess the implications of these different formulae to simulating secondary production.

*To further address this point we now have the following text "Uncertainties in model grazing formulations could also contribute to model-data mismatch (Gentleman et al., 2003a; Sailley et al., 2015). Future in situ grazing measurements are needed to enable an objective selection of grazing formulations and parameter values. In particular, field studies that shed light on prey selectivity would be useful for parameterizing PBMs with multiple mesozooplankton functional groups, such as NEMURO-GoM. Such studies are challenging, however, because the difficulty of making in situ grazing measurements on mesozooplankton, combined with the inherent uncertainty of these measurements can make it challenging to differentiate between, for instance Ivlev and Holling's disk grazing formulations (e.g., Fig. 4 in Morrow et al. (2018)). Nevertheless, differences in parameterizing grazing can lead to substantially different model behavior (Anderson et al., 2010; Sailley et al., 2015; Wainwright et al., 2007). In NEMURO-GoM, secondary production and dietary preferences of the mesozooplankton community are both strongly influenced by model grazing formulation. While we carefully chose parameterizations that gave reasonable fits to extensive field datasets of zooplankton biomass and grazing rates, this does not preclude the possibility that other functional forms would have more accurately simulated zooplankton dynamics. Hence, future PBMs should investigate how different grazing formulations impact zooplankton dynamics in the region. We especially recommend collaborations between experimentalists (potentially using new techniques such as DNA metabarcoding of gut contents) and modelers to develop synthetic approaches with the potential to quantitatively assess the realism of different grazing formulations."(lines 894-912).*

The model greatly overestimates chlorophyll in the coastal region, but the authors did not recognize this problem… How does affect the simulated zooplankton biomass and dynamics? Maybe they are having the right spatial patterns for zooplankton biomass but wrong underlying dynamics.

*We respectfully disagree with Referee #2 in regards to not recognizing that the model overestimates Chl in the coastal region. To draw attention to this issue we state "While a clear shelf signature is well resolved in NEMURO-GoM, the model-data mismatch is greater on the shelf compared to oligotrophic region"(lines 761-762) and "In our model, the most noticeable surface Chl model-data mismatch occurs on the southern GoM shelf (Campeche Bank (CB)), where the model consistently overestimates surface Chl" (lines 787-788). We also note that we intentionally included Fig. 2F to draw readers' attention to our models imperfect simulation of coastal Chl. We intentionally included model-data validation plots that highlighted the weaknesses of our model, rather than only showing the strengths. In the revised manuscript we have also included a specific section in our Discussion section dedicated to addressing the issue of surface chlorophyll discrepancies in the model.*

*We also believe that the reviewer is overstating the model-data mismatch on the shelf when he/she states that our model "greatly overestimates chlorophyll in the coastal region". Many other models of the GoM also struggle to accurately capture variability in phytoplankton biomass on the shelf. We believe we have made a significant step forward in this regard by developing a model that reproduces the dominant spatial patterns on the shelf. Our model even accurately resolves the small shelf signature in the western GoM. On average, the model typically overestimates Chl by anywhere from a very slight overestimate to a factor of 2 overestimates (and occasionally a factor of 3 overestimate). This is not a huge discrepancy compared to other GoM models (see, for instance Fig. 3 of Gomez et al. 2018 which shows that their model consistently underestimates phytoplankton biomass in the Mississippi Delta and Texas Shelf regions, with almost 5-fold underestimates common in the summer months). Furthermore, as we note, the shelf model-data discrepancy is mostly driven by a substantial overestimate of Chl on the Campeche Bank. Overall, we actually consider our model to have relatively decent agreement with surface chl measurements on the shelf, considering that it was a model specifically focused on open ocean regions. Nevertheless, there are clear areas for improvement on the shelf (e.g. more realistic C:Chl models and including (non-climatological) river nutrient data). We have highlighted these in the revised manuscript (lines 784-797).*

The authors should include a good validation for nutrients and primary production in the coastal region; otherwise, we do not know if the model is reproducing well cross-shore patterns. The authors should also include a validation for phytoplankton structure (e.g., diatom to total phytoplankton ratio).

*We do not believe that substantial validation of coastal region nutrients, primary production, and phytoplankton structure in the coastal region is justified in this (already long) manuscript for two reasons:*

*1) Our stated focus is on zooplankton dynamics in the oligotrophic GoM, rather than on phytoplankton dynamics in the coastal region.*

*2) There is a paucity of* in situ *data to compare the model to: Spatiotemporal patterns in nutrients and primary production are highly variable in the coastal region, and unfortunately the publicly available* in situ *measurements (at least that we have access to) are not sufficient to resolve this spatiotemporal variability sufficiently to enable detailed quantitative comparison. This does not mean that we have not considered such things as nutrient concentrations, primary productivity, and SP:LP ratios in our model. For instance in the supplement, we noted that LP:SP ratios are an important metric for evaluating the simulated ecosystem which is why we considered this metric during the model tuning process. We have now added an LP:SP ratio map to the supplemental material, so that the reviewer can see this metric. However, there are few true* in situ *measurements of LP:SP biomass ratio in the Gulf to use as true "validation" rather than a subjective validation metric. We hope that future in situ research programs will produce more of such measurements, or alternately, that the planned NASA PACE project (and accompanying* in situ *validation studies) will enable accurate determination of phytoplankton taxonomic groups from satellite.*